

# Transcription factors involved in abiotic stress responses in Maize (*Zea mays* L.) and their roles in enhanced productivity in the post genomics era

Roy Njoroge Kimotho[1,2], Elamin Hafiz Baillo[1,2] and Zhengbin Zhang[1,2,3]

[1] Key Laboratory of Agricultural Water Resources, Hebei Laboratory of Agricultural Water Saving, Center for Agricultural Resources Research, Institute of Genetics and Developmental Biology, Chinese Academy of Sciences, Shijiazhuang, Hebei, China
[2] University of Chinese Academy of Sciences, Beijing, China
[3] Innovative Academy of Seed Design, Chinese Academy of Sciences, Beijing, China

Corresponding author
Zhengbin Zhang, zzb@sjziam.ac.cn

## ABSTRACT

**Background:** Maize (*Zea mays* L.) is a principal cereal crop cultivated worldwide for human food, animal feed, and more recently as a source of biofuel. However, as a direct consequence of water insufficiency and climate change, frequent occurrences of both biotic and abiotic stresses have been reported in various regions around the world, and recently, this has become a constant threat in increasing global maize yields. Plants respond to abiotic stresses by utilizing the activities of transcription factors (TFs), which are families of genes coding for specific TF proteins. TF target genes form a regulon that is involved in the repression/activation of genes associated with abiotic stress responses. Therefore, it is of utmost importance to have a systematic study on each TF family, the downstream target genes they regulate, and the specific TF genes involved in multiple abiotic stress responses in maize and other staple crops.

**Method:** In this review, the main TF families, the specific TF genes and their regulons that are involved in abiotic stress regulation will be briefly discussed. Great emphasis will be given on maize abiotic stress improvement throughout this review, although other examples from different plants like rice, Arabidopsis, wheat, and barley will be used.

**Results:** We have described in detail the main TF families in maize that take part in abiotic stress responses together with their regulons. Furthermore, we have also briefly described the utilization of high-efficiency technologies in the study and characterization of TFs involved in the abiotic stress regulatory networks in plants with an emphasis on increasing maize production. Examples of these technologies include next-generation sequencing, microarray analysis, machine learning, and RNA-Seq.

**Conclusion:** In conclusion, it is expected that all the information provided in this review will in time contribute to the use of TF genes in the research, breeding, and development of new abiotic stress tolerant maize cultivars.

## INTRODUCTION

Abiotic stresses, for instance drought, salinity, cold, high temperatures, and mineral toxicity, are the main cause of major crop yield reductions worldwide, reducing expected average yields of the major crops by more than 50% (*Prasad et al., 2011*; *Mahalingam, 2015*). Plants are sessile beings that are continuously exposed to various changes in the environmental conditions. Variations in the environment involving both biotic and abiotic stresses have negative effects on economically important crops like maize (*Ramegowda & Senthil-Kumar, 2015*). Evolutionary changes have helped many plants adapt to different adverse conditions. Some species show a marked increase in tolerance to various abiotic stresses compared to others (*Phukan et al., 2014*). Due to global warming and the climatic abnormalities accompanying it, the frequencies of combined biotic and abiotic stresses have significantly increased, leading to reduced growth and yield of the major crops worldwide (*Mittler, 2006*; *Pandey, Ramegowda & Senthil-Kumar, 2015*; *Ramegowda & Senthil-Kumar, 2015*). Moreover, continuous manifestations of abiotic stresses such as heat and drought together, has led to severe reductions in crop yields as opposed to when these stresses occur separately during the different growth stages (*Mittler, 2006*; *Prasad et al., 2011*).

Maize (*Zea mays* ssp. Mays L.) is one of the most important cereal crops cultivated worldwide (particularly in Africa and South America). Global maize production increased from 255 million tons in 1968 to 1,134 million tons in 2017 representing an average annual growth of 3.46% (https://knoema.com/atlas/World/topics/Agriculture/Crops-Production-Quantity-tonnes/Maize-production). Maize production has significantly increased in both the developing countries and the developed countries (*Wang et al., 2013*).

Maize is a staple food in many parts of the world; it is consumed directly by humans, used for animal feed, and in other maize products such as corn syrup and corn starch. In the last century, maize has been utilized as a model system in the study of various biological events and systems including paramutation, transposition, allelic diversity, and heterosis (*Bennetzen & Hake, 2009*). Recently, maize has been identified as a potential sustainable feedstock, as well as a model system for research in the bioenergy and biofuel industries (*Perlack et al., 2005*). Continuous studies in maize genetics has led to further understanding of other related C4 grasses such as Elephant grass (*Miscanthus gigantus*) and switchgrass (*Panicum virgatum*) as scientists aim to develop and domesticate these plants (*Perlack et al., 2005*). However, maize belts around the world which range from the latitude 40° South to the latitude 58° North are exposed to continuous effects of both biotic and abiotic stresses (*Gong et al., 2014*). Abiotic stresses, like salinity, drought, nutrient deficiency, and high and low temperatures are the major environmental factors that negatively influence maize production. In particular, intense waterlogging, extreme temperatures, and drought have significantly affected maize yields (*Ahuja et al., 2010*).

Plants must cope with a variety of abiotic stresses including extreme temperatures, heavy metals, osmotic stresses, and high light intensity. Under stress, accumulation of some metabolites positively regulates plants response to both abiotic and biotic stresses thus protecting plants from multiple stresses (*Rasmussen et al., 2013*; *Suzuki et al., 2014*).
Changes in ions fluxes, callose accumulation, phytohormones, and reactive oxygen species (ROS) are the first responses induced to tackle the stresses, leading to metabolic reprogramming in the plants defenses (*Bartoli et al., 2013*).

Reactive oxygen species such as hydrogen peroxide ($H_2O_2$) and superoxide ($\bullet O{-}_2$) which are produced due to oxidative stresses, inhibit photosynthesis and cause vast cellular destruction (*Allan & Fluhr, 2007*). ROS are normally removed rapidly by antioxidative mechanisms, although this removal can be hindered by the stresses leading to an increase in ROS concentration inside the cells, and further increasing the damage caused (*Allan & Fluhr, 2007*).

Another pathway involved in abiotic stress responses in plants is the mitogen-activated protein kinase (MAPK) cascades. MAPK cascades are activated following the recognition and perception of stress stimuli and control the stress response pathways (*Wurzinger et al., 2011*). They are highly conserved in eukaryotes and are responsible for signal transduction in various cellular processes under different biotic and abiotic stress responses. Because MAPKs are involved in various stress responses, they play a main role in the combination of biotic and abiotic stresses (*Amajová et al., 2013*).

Additionally, hormone signaling in plants is another important pathway involved in biotic and abiotic stress responses and the primary hormone involved is abscisic acid (ABA). An increase in ABA concentration in plants under abiotic stress modulates the abiotic stress-regulation network (*Xiong, Schumaker & Zhu, 2002*), while biotic stress is mediated by antagonism in other stress hormones such as jasmonic-acid (JA)/ethylene (ET) and salicylic acid (SA; *Liu et al., 2008*). The role of ABA in abiotic stress responses has been widely described throughout this review.

Plants do not respond to multiple stresses by way of a linear pathway. The responses are complex circuits involving various pathways in tissues, cellular specific compartments, and the interactions of signaling molecules in controlling a particular response to a stimulus (*Dombrowski, 2003*). Due to abiotic stresses, numerous proteins, and gene transcripts are altered through the regulation of protein turnover and gene expression (*Jiang et al., 2007*; *Wong et al., 2006*).

In this review, we will briefly describe the main transcription factor (TF) families and the interactions of these TFs with the *cis*-acting elements (CREs) which are present in the promoter regions of stress responsive genes. Even though TF regulons have been described recently by (*Gahlaut et al., 2016*; *Joshi et al., 2016*), this review will focus on TFs involved in abiotic stress tolerance with a specific focus on maize. We will also focus on new ways of increasing production of maize by utilizing currently available genomic information, tools, and data.

## Survey methodology

All published manuscripts cited in this review were obtained from different databases including Pubmed, Web of Science, EBSCO, Google Scholar, research gate, Science Direct, SCOPUS, JSTOR, SciELO, and Semantic Scholar. Keywords such as "maize stress tolerance," "TFs involved in abiotic stress responses," "abiotic stress," "TF downstream genes," and "regulons involved in abiotic stress" were searched between January 10 and

October 25, 2018. We have critically analyzed articles to provide an in-depth and comprehensive research trend focusing on the TFs involved in abiotic stress tolerance in maize. Furthermore, we have provided perspectives on the latest research as well as previous findings with focus on TF families involved in abiotic stress responses in maize.

## Functions of transcription factors

Abiotic stress-induced genes are generally classified into two main groups based on their protein products. One type includes the genes coding for products which directly allow cells to resist environmental stresses for instance osmotic regulatory protein, late embryogenesis abundant (LEA) protein, enzymes synthesizing proline, betamine, malondialdehyde (MDA), and other osmotic regulators and anti-freezing proteins (*Ciarmiello et al., 2011*). The second type of genes are regulatory proteins which operate in the signal transduction networks, for example, molecular chaperones, functional proteins, and TFs or kinases (*Song, Li & Hou, 2013*; *Ciarmiello et al., 2011*).

Networks of TFs together with transcription factor binding sites (TFBS) directly control transcriptional regulation of plant genes (*Chaves & Oliveira, 2004*). TFs are proteins usually consisting of two domains, namely (1) the DNA binding domain (DB) and (2) an activation domain (AD). A TF binds to the *cis*-acting element (TF binding site) located in the promoter region of a stress-induced gene with the support of a DB domain (*Yamasaki et al., 2013*). This event brings the AD close to the target gene leading to repression or activation of this gene. A large percentage of genes in the plant genome (nearly 10%) essentially encode for TFs (*Franco-Zorrilla et al., 2014*). TFs activate or repress the activity of RNA polymerase, leading to gene regulation. TFs can be categorized into various families based on their DNA DBs (*Riechmann et al., 2000*). Since abiotic stresses are quantitative traits that might require regulation of several genes including the TF genes, and since a single TF may regulate several genes that are involved in abiotic stress tolerance. A detailed study of all TFs associated with abiotic stress regulatory mechanisms in maize will be significantly rewarding. For example, *Xu et al. (2006)* successfully converted flood sensitive rice genotypes into flood-tolerant varieties by introgression of the *sub1* locus which encodes an (ET response factor) TF, leading to the induction of about 900 stress-responsive genes.

Transcription factor DNA-DBs are strongly conserved between species, to the extent of using these characteristics to classify the TFs into various families (Fig. 1). These families differ among plant species in that different plant systems have between 26 and 83 TFs families (*Jin et al., 2014*). In *Arabidopsis*, for instance, approximately 34 families consisting of 1,533 TFs have so far been classified (*Riechmann et al., 2000*). Additionally, in *Arabidopsis* and many other plants, transcriptome data revealed a number of pathways which respond to abiotic stresses independently, pointing to the possibility that susceptibility or tolerance of both biotic and abiotic stresses are controlled by a sophisticated gene regulatory network (GRN) at the transcriptome level (*Umezawa et al., 2006*).

Abscisic acid is the principal hormone involved in the coordination of abiotic stress in plants (Fig. 1). This hormone regulates a complex gene regulatory system that enables

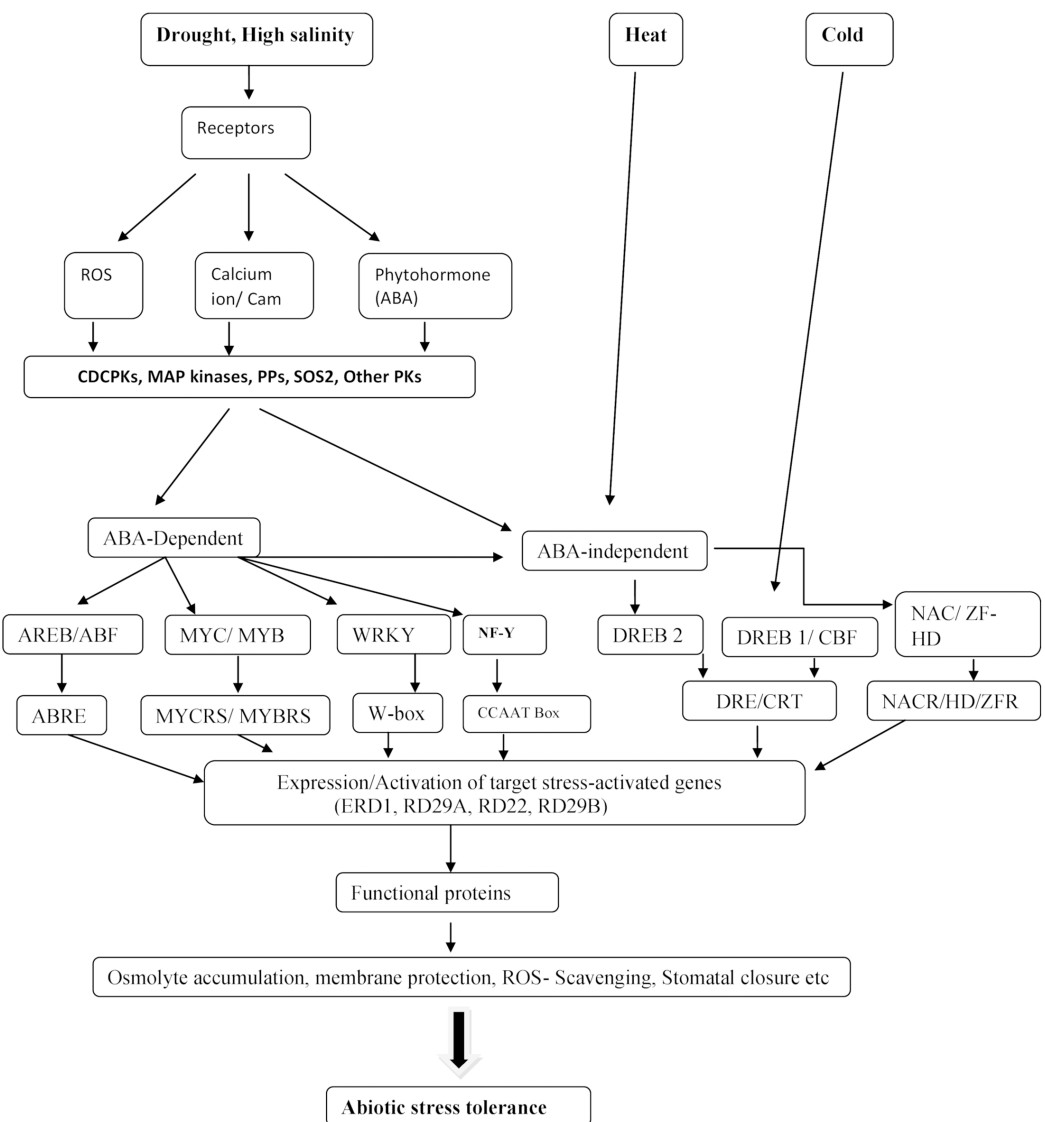

**Figure 1 Gene expression and abiotic stress signal perception in plants.** A diagrammatic representation of gene expression and abiotic stress signal perception in plants via ABA-independent and ABA-dependent pathways (modified from *Gahlaut et al., 2016*; *Khan et al., 2018*). Abbreviations: Abscisic acid (ABA), Reactive oxygen species (ROS), myeloblastosis oncogene (MYB), myelocytomatosis oncogene (MYC), Zinc-finger homeodomain (ZF-HD) regulon, ABA-responsive element binding protein (AREB), ABA-independent regulons include the NAC (CUC, NAM, and ATAF), The *cis*-acting element (DRE), ABA-binding factor (ABF), The *cis*-acting element (CRE), Dehydration responsive element binding proteins (DREBs), C-repeat (CRT), (ZFR) zinc finger RNA binding protein, (NARC) NAC recognition site, (MYBRS) MYB recognition site, (MYCRS) MYC recognition site, Nuclear transcription factor Y (NF-Y), Heat Shock Factors (HSFs), Inducer of CBF Expression (ICE).

plants to handle decreased moisture availability (*Cutler et al., 2010*). ABA-dependent gene activation pathways were identified as pathways which determine stress tolerance by the induction of at least two separate regulons: the first one is the myeloblastosis oncogene (MYB)/myelocytomatosis oncogene (MYC) regulon, and the second one is the

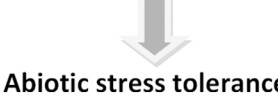

**Figure 2  Cross-talk network among *cis*-acting elements and transcription factors.** Cross-talk network between *cis*-acting elements and TFs in the ABA-independent and ABA-dependent pathways during abiotic stress. Broken arrows indicate the protein-protein interactions. Thick green arrows show the major pathways which regulate many downstream genes (modified from *Yamaguchi-Shinozaki & Shinozaki, 2006*).

ABA-responsive element binding protein/ABA-binding factor (AREB/ABF) (*Abe et al., 1997*; *Saibo, Lourenço & Oliveira, 2009*) (Fig. 1). ABA-independent regulons include the NAC (CUC, NAM, and ATAF) and the zinc-finger homeodomain (HD) regulon (*Nakashima, Ito & Yamaguchi-Shinozaki, 2009*; *Saibo, Lourenço & Oliveira, 2009*) (Fig. 1).

The different stress tolerance responsive TFs normally function independently, although there is a possibility that some level of cross-link occurs between these TFs (Fig. 2). Many studies have shown that ABA-independent and ABA-dependent pathways might converge at several unexpected points. This points of convergence represent transcriptional repressors and enhancers interacting indirectly or directly with DRE/C repeat and ABA-responsive element (ABRE) and hence initiate synergistic interactions between the ABA response and osmotic stress (Fig. 2).

The above mentioned TF families have been studied in detail in a number of important food crops and also in model plant systems including; *Arabidopsis thaliana*, *Oryza sativa*, *Triticum aestivum*, *Sorghum bicolor*, *Vitis vinifera*, *Hordeum vulgare*,

*Solanum tuberosum*, and *Brassica napus*. Recent studies have shown the functions of abiotic stress-responsive TFs, and their potential roles to be used in future for purposes of molecular breeding and improvement of different crop varieties.

Much progress has been achieved in our understanding of transcriptional regulation, signal transduction, and gene expression in plant responses to abiotic stresses (*Zhu et al., 2010*). In rice, for example, overexpression of a NAC TF encoding gene, *SNAC1*, resulted in increased yields and increased tolerance to drought in transgenic plants (*Hu et al., 2006*). Overexpression of a *Glycine soja* NAC TF designated as *GsNAC019* in transgenic Arabidopsis resulted in plants that were tolerant to alkaline stress at both the seedling and mature stages although the transgenic plants had reduced sensitivity to ABA (*Cao et al., 2017*). Similarly, functional analysis of a *Pyrus betulifolia* NAC TF gene designated as *PbeNAC1*, revealed that this gene is involved in the regulation of cold and drought stress tolerance (*Jin et al., 2017*). Additionally, a chickpea (*Cicer arietinum*) stress associated TF, *CarNAC4* was linked with reduced MDA content and water stress rates in response to salinity and drought stress respectively (*Yu et al., 2016*).

*Ramakrishna et al. (2018)* showed that overexpression of a finger millet bZIP TF gene *EcbZIP17* in tobacco plants resulted in higher germination rate, increased biomass, and increased survival rate in the transgenic plants. Furthermore, the transgenic tobacco plants also showed increased seed yields compared to the control plants. Likewise, *Xu et al. (2006)* showed that transgenic Arabidopsis and soybean seedlings overexpressing a soybean bZIP TF designated as *GmbZIP110* had improved salt tolerance, suggesting that *GmbZIP110* functions as a positive regulator involved in salt stress tolerance. Functional analysis of *GmbZIP110* in transgenic Arabidopsis revealed that this gene could bind to the ACGT motif and regulate many downstream target genes (*Cao et al., 2017*). Elsewhere, overexpression of an Arabidopsis bZIP TF designated as *ABF3* in transgenic alfalfa (*Medicago sativa*) under the command of a sweet potato oxidative stress-inducible promoter *SWPA2*, resulted in improved growth under drought stress (*Wang et al., 2016c*). In hot pepper (*Capsicum annuum*), overexpression of a bZIP encoding gene *CaBZ1* in transgenic potato significantly improved dehydration stress tolerance without any detrimental effects on plant growth or yield (*Moon et al., 2015*).

Overexpression of *OsMYB55*, a rice MYB encoding gene in transgenic maize resulted in improved plant growth as well as decreased negative effects of drought and high temperature (*Casaretto et al., 2016*). *Wei et al. (2017)* demonstrated that *CiMYB3* and *CiMYB5* cloned from *Cichorium intybus* were both involved in the fructan pathway degradation in response to various abiotic stresses. In banana (*Musa paradisiaca*), overexpression of an MYB TF gene designated as *MpMYBS3* significantly improved tolerance to cold stress in transgenic plants (*Dou et al., 2015*). Elsewhere, a MYB TF gene designated as *MtMYBS* from *Medicago truncatula* was able to enhance salt and drought tolerance in transgenic Arabidopsis by improving the primary root growth (*Dong et al., 2017*). Likewise, overexpression of *GaMYB62L* from cotton (*Gossypium arboreum*) in transgenic *Arabidopsis* resulted in enhanced drought tolerance (*Butt et al., 2017*).

The exogenous expression of *AtDREB1A* (dehydration responsive element binding proteins (DREB)) gene from *Arabidopsis* in transgenic *Salvia miltiorrhiza* resulted in

plants with higher antioxidant activities and photosynthetic rates under drought stress (*Wei et al., 2016*). Elsewhere, overexpression of *SbDREB2A* from *Salicornia brachita* in transgenic tobacco resulted in improved growth and seed germination under hyperionic and hyperosmotic stresses (*Gupta, Jha & Agarwal, 2014*). *Zhang et al. (2015)* cloned SsDREB protein from *Suaeda salsa* and showed that this protein enhances the photosynthesis rate in transgenic tobacco plants under drought and salt stresses.

In the WYKY TF gene family, *OsWRKY71* from rice was found to act as a positive regulator to cold stress tolerance by regulating several downstream genes like *WSI76* and *OsTGFR* (*Kim et al., 2016*). Virus-induced gene silencing of the *GhWRKY6* gene from cotton (*Gossypium hirsutum*) led to increased sensitivity to various abiotic stresses in the silenced plants (*Ullah et al., 2017*). Elsewhere, *SIDRW1* and *SLWRKY39* which are WRKY TFs were found to confer both abiotic and biotic stress tolerance in tomato (*Solanum lycopersicum*) by activating both abiotic stress and pathogenesis-related downstream genes (*Liu et al., 2014a*; *Sun et al., 2015*).

Transcription factors and regulons involved in abiotic stress regulation from other TF families have also been identified and described. For instance, in *Populus euphratica*, exogenous expression of *PeHLH35* belonging to the bHLH TF family resulted in significant improvement in water deficit tolerance through changes in several physiological processes such as stomatal density and transcription rate (*Dong et al., 2014*). In tomato, overexpression of a cycling Dof factor (CDF) TF designated as *CDF3* resulted in increased biomass production and higher yields in transgenic tomato plants under salt stress (*Renau-Morata et al., 2017*).

## TFs and the specific target genes involved in abiotic stress tolerance in maize

### MYC/MYB regulon

The MYC/MYB families of TF proteins have diverse functions and are found in both animals and plants (*Abe et al., 2003*). Both MYB/MYC TFs participate in the ABA-dependent pathway involved in abiotic stress signaling (Fig. 1). The first MYB TF gene in plants was identified in Maize and was designated as *C1*, it codes for a c-MYB like TF that is involved in the biosynthesis of anthocyanin (*Paz-Ares et al., 1987*). In the MYB family, each TF consists of an MYB domain containing 1–3 imperfect repeats and is made up of around 52 amino acid residues with a helix-turn-helix conformation which interposes inside the major grooves of DNA (*Yanhui et al., 2006*). Both MYB and MYC TFs are usually involved in making up the common regulons known as the MYB/MYC regulons (*Gahlaut et al., 2016*).

In the maize genome, *Du et al. (2013)* reported 72 MYB related proteins. *Chen et al. (2017)* analyzed the expression profiles of 46 MYB genes from maize, in response to various abiotic stresses and found 22 genes that responded to the different stress conditions. Additionally, 16 of these genes were induced in response to a minimum of two stresses. These results suggesting that these genes could take part in signal transduction pathways involved in abiotic stress responses. Additionally, the function of *ZmMYB30* which was significantly up-regulated under drought, salt, and ABA stresses was further analyzed (*Chen et al., 2017*)

**Table 1 Abiotic stress-related TF families, together with the specific TFs in Maize.**

| Family | TFs in maize | Cis-element recognition | Stress response | Downstream genes | References |
|---|---|---|---|---|---|
| DREB/CBF | ZmDREB2A | (DRE) TACCGACAT | Salt, heat, drought, cold | rd29A, rd29B, ZmGOLS2 | Qin et al. (2007) |
| | ZmDBP3 | (DRE) TACCGACAT | Cold, salt | U | Wang & Dong (2009) and Chang & Yin (2009) |
| | ZmDREB1A | (DRE/CRT) G/ACCGAC | Drought, cold | KIN1, KIN2, COR15A, etc | Qin et al. (2004) |
| | ZmDBF3 | N/A | Salt, drought, freezing | U | Zhou et al. (2016) |
| | ZmDBP4 | (DRE/CRT) G/ACCGAC | Cold, drought | U | Wang, Yang & Yang (2011) |
| | ZmDREB2.7 | (DRE) A/ GCCGAC | Drought | U | Liu et al. (2013) |
| MYB/MYC | ZmMYB30 | (MYBR) TAACNA/G | Salt, drought, ABA | RD20, RD29A, RbohD, etc | Chen et al. (2017) |
| | ZmMYB36 | N/A | Salt, drought, ABA | U | Chen et al. (2017) |
| | ZmMYB95 | N/A | Salt, drought, ABA | U | Chen et al. (2017) |
| | ZmMYB53 | N/A | Cold | U | Chen et al. (2017) |
| | ZmMYB31 | N/A | Sensitivity to UV radiation | ZmF5H, ZmCOMT, C3H, and ZmActin | Fornalé et al. (2010) |
| | ZmMYB-R1 | N/A | Cold, high salinity, drought, ABA, and heat | U | Liu et al. (2012) |
| bZIP | ZmbZIP60 | (ABRE) ACGTGGC | Dehydration, high salinity, ABA | U | Wang et al. (2012) |
| | | | Heat stress | U | Li et al. (2018) |
| | ZmbZIP17 | N/A | Drought, ABA, heat, salt | U | Jia et al. (2009) |
| | ZmbZIP54 and ZmbZIP107 | (ABRE) ACGTGGC | Lead (Pb) | U | Zhang et al. (2017a) |
| | mlip15 | (ABRE) ACGTGGC | Low temperature, salt, ABA | U | Kusano et al. (1995) |
| | ZmbZIP72 | (ABRE) ACGTGGC | ABA, drought, high salinity | Rab18, rd29B, HIS1-3, etc | Ying et al. (2012) |
| | ABP9 | (ABRE) (C/T) ACGTGGC | ABA, drought, $H_2O_2$, salt | KIN1, COR15A, PP2C, AZF2, etc | Zhang et al. (2011) |
| | ZmbZIP4 | (ABRE) (C/T) ACGTGGC | Heat, cold, salinity, and ABA | ZmLEA2, ZmRD20, ZMRab18, ZmGEA6, etc | Ma et al. (2018) |
| NAC | ZmSNAC1 | N/A | Low temperature, ABA, high salinity, drought | U | Lu et al. (2012) |
| | ZmNAC55 | N/A | High salinity, cold, drought, ABA | RD29B, LEA14, RD17, ZAT10, etc | Mao et al. (2016) |
| | Zma006493 | N/A | Drought | U | Lu et al. (2015) |
| | Zma000584 | N/A | Drought, cold | U | Lu et al. (2015) |
| | Zma001259 | N/A | Drought, salt, cold | U | Lu et al. (2015) |
| | ZmSNAC052 | N/A | Drought, cold | U | Lu et al. (2015) |
| | Zma029150 | N/A | Drought, salt | U | Lu et al. (2015) |

| Family | TFs in maize | Cis-element recognition | Stress response | Downstream genes | References |
|---|---|---|---|---|---|
| WRKY | ZmWRKY17 | (W-box) TTGACC/T | Drought, salt, ABA | bHLH92, KIN1, DREB1F, etc | Cai et al. (2017) |
| | ZmWRKY33 | (W-box) TTGACC/T | High salinity, dehydration, cold, ABA | RD29A and DREB1B | Li et al. (2013) |
| | ZmWRKY44 | (W-box) TTGACC/T | Salt, high temperature, ABA, $H_2O_2$ | U | Wang (2014) |
| | ZmWRKY58 | (W-box) TTGACC/T | Drought, ABA, salt | U | Cai et al. (2014) |
| | ZmWRKY106 | (W-box) TTGACC/T | Drought, high temperature, ABA, salt | CuZnSOD, DREB2A, NCED6, and RD29A | Wang et al. (2018c) |
| | ZmWRKY40 | (W-box) TTGACC/T | Drought, high salinity, high temperature, ABA | DREB2B, STZ, and RD29A | Wang et al. (2018a) |
| Others | | | | | |
| HD-Zip | Zmhdz10 | CAATAATTG | Salt, ABA | ABI1, RD22, P5CS1, etc | Zhao et al. (2018) |
| | ZmHDZ4 | CAATAATTG | Drought | U | Wu et al. (2016) |
| HSP | ZmERD2 | N/A | Heat, salinity, cold, PEG, dehydration | U | Song et al. (2016) |
| NF-Y | ZmNF-YB16 | CCAAT | Dehydration, Drought | P5CS, Atj3, AtDJC82, HSP70, etc | Wang et al. (2018b) |
| | ZmNF-YB2 | CCAAT | Drought | U | Nelson et al. (2007) |
| | ZmNF-YA3 | CCAAT | Drought, high temperature | ZmbHLH92, ZmMYC4, and ZmFAMA | Su et al. (2018) |

**Notes:**
Abiotic stress-related TF families, together with the specific TFs, their characteristics, the regulons they control and their regulatory functions in the abiotic stress responses in Maize.
N/A-The cis-acting element is unknown, U-unknown.

(Table 1). Exogenous expression of *ZmMYB30* in *Arabidopsis* stimulated tolerance to salt and elevated the expression of eight abiotic stress corresponding genes (*ABF3, ATGolS2, AB15, DREB2A, RD20, RD29B, RD29A,* and *MYB2*) enabling transgenic plants to be more tolerant to adverse environmental conditions (*Chen et al., 2017*) (Table 1). Moreover, another six genes (*RD22, RbohD, P5CS1, RAB18, RbohF,* and *LEA14*) were either unchanged or slightly elevated in the transgenic *Arabidopsis* plants.

Another maize MYB TF, *ZmMYB31* was found to repress the biosynthesis of sinopoylmalate leading to increased sensitivity to UV irradiation and dwarfism in transgenic plants (*Fornalé et al., 2010*). Furthermore, *ZmMYB31* activated a number of stress-responsive genes (*ZmF5H, C3H, ZmActin,* and *ZmCOMT*) in vivo in maize, and *4CL1* and *COMT* genes in transgenic *Arabidopsis*. The roles of maize MYB related genes in response to drought stress were examined based on microarray data (*Dash et al., 2012; Du et al., 2013*). On the maize 18k GeneChip, 26 probe sets were shown to correspond to 32 MYB-related genes (whereby five probes represented one gene). Further analysis of this highly similar sequence data revealed that the majority of the MYB-related genes were expressed at low levels, although their expression was in response to a specific stress. Elsewhere, gene expression analysis between two maize varieties, a drought sensitive (Ye478) variety, and a drought tolerant (Han21) variety was found to be very similar (*Du et al., 2013*). For instance, four

CCA1-like/R–R genes (*ZmMYBR49, ZmMYBR19, ZmMYBR56,* and *ZmMYBR28*), six TBP-like genes (*ZmMYBR55, ZmMYBR45, ZmMYBR47, ZmMYBR31, ZmMYBR26,* and *ZmMYBR07*) and a single TRF-like gene (*ZmMYBR41*) were all elevated in response to drought stress. Expression analysis of *ZmMYBR37* an I-box-like gene, and five CCA1-like/R–R genes (*ZmMYBR63, ZmMYBR44, ZmMYBR27, ZmMYBR18,* and *ZmMYBR03*), showed that these genes were highly down-regulated in response to drought stress. Although recovery of all these genes above was observed after re-watering (Table 1). Similarly, a maize R1-type TF that is encoded by *ZmMYB-R1* gene was activated by cold, exogenous ABA, drought, heat and high salinity (*Liu et al., 2012*). Functional analysis of *ZmMYB-R1* in different tissues revealed it first reached its maximum levels in the leaves and later it was detected in the roots and stems.

In the model plant *Arabidopsis*, MYB and MYC TFs were found to accumulate in plant tissues following the accumulation of ABA (*Lata, Yadav & Prasad, 2007*). Seven *Arabidopsis* MYB TF genes namely *AtMYBCDC5, AtMYB77, AtMYB73, AtMYB44, AtMYB6, AtMYB7,* and *AtMYB4* were all strongly expressed in all organs, in response to several abiotic stresses (*Yanhui et al., 2006*). Functional analysis of two MYB/MYC genes (*AtMYC2* and *AtMYB2*), in transgenic *Arabidopsis* revealed that the TF proteins encoded by these two genes can bind to the promoter regions of several ABA or JA inducible genes. For example, *AtADH1* and *RD22* thus making the transgenic *Arabidopsis* plants ABA-responsive and more tolerant to both drought and osmotic stress (*Abe et al., 2003*).

Taken together, all the above findings suggest that *MYB* genes could be engineered in crops leading to activation of general pathways involved in abiotic stress responses in plants. For instance, overexpression of *OsMYB55*, a rice R2R3-MYB TF significantly improved rice plants tolerance to extreme temperatures, and this was directly linked with improved amino acid metabolism (*El-kereamy et al., 2012*). Additionally, these findings will facilitate our understanding of gene regulation by MYB TFs, thus leading to the development of new abiotic stress tolerant crop varieties. Finally, phylogenetic, functional, and structural analyses revealed that most homologous MYB proteins that possess conserved domains have similar activities and functions in divergent plant species. Although a lot of information is available on the molecular functioning of MYB TFs in abiotic stress responses, deciphering the downstream and upstream events in MYB TFs in abiotic stress responses remains an immense undertaking.

### The AP2/EREBP regulons

The AP2/EREBP (ET-responsive element binding protein) family is made up of a large group of plant specific TFs that are characterized by the presence of a highly conserved AP2/ET-responsive element binding factor (ERF). The ERF interacts directly with GCC box and/or DRE/C-repeat element (CRT) at the promoter site of downstream target genes (*Riechmann & Meyerowitz, 1998*). AP2/EREBP TFs play vital roles in stress responses and developmental processes such as cell proliferation, plant hormone responses, and biotic and abiotic stress responses (*Sharoni et al., 2011*). Based on the similarity and number of AP2/ERF domains, AP2/EREBP TFs are grouped into four main subfamilies: ERF, RAV (related to AB13/VP1), DREB and AP2 (Apetala 2) (*Sharoni et al., 2011*). Among these four,

the DREB subfamily has been extensively studied due to the roles its TFs play in plant biotic and abiotic stress responses.

Dehydration responsive element binding proteins play a significant role in the ABA-independent pathways that are responsible for the activation of abiotic stress-regulatory genes (*Lata, Yadav & Prasad, 2007*). DREB TFs are made up of one AP2/ERF DNA binding region, which binds to the *cis*-acting element DRE composed of TACGACAT (a nine bp core sequence) and is present in the promoters of abiotic stress-responsive genes (*Gahlaut et al., 2016*). The existence of this *cis*-acting element (CRE) has been recorded in several abiotic stress-responsive genes, for example, *RD29B* and *RD29A* in *Arabidopsis* (*Yamaguchi-Shinozaki & Shinozaki, 1994*). CRT *cis*-acting element consisting of the A/GCCGAC motif and which are similar to DREBs, have been identified in the promoter regions of cold-responsive genes in *Arabidopsis*, whereby the CRT elements bind to the TF CBF (CRT binding factor) in response to cold stress (*Saleh, Lumreras & Pages, 2005*). In *Arabidopsis*, for example, exogenous overexpression of *AtDREB1/CBF* led to cold, drought, and high salinity tolerance in transgenic plants. These findings clearly suggest that DREBs/CBFs TFs have the potential to target multiple stress-responsive candidate genes in the plant genomes (*Jaglo-Ottosen et al., 1998*; *Kasuga et al., 1999*).

In maize, the role of DREB TFs has been investigated by adopting both molecular and genetic analyses (*Zhuang et al., 2010*). For example, *ZmDREB1A* was activated by cold stress and moderately elevated by high-salinity stress in maize seedlings (*Qin et al., 2004*) (Table 1). Overexpression of *ZmDREB1A* in transgenic *Arabidopsis* led to induced overexpression of abiotic stress-activated genes giving rise to plants with enhanced tolerance to extreme drought and freezing stresses (Table 2). Investigations were done to ascertain whether *ZmDREB1A* could induce other genes in the dehydration and/or cold pathways of wild-type plants. The results revealed that expression levels of *KIN1*, *KIN2*, and *COR15A* were all highly up-regulated in the 35S:ZmDREB1Aa transgenic line under normal conditions when compared to the wild-type plants. Expression analysis of *RD17, ERD10*, and *RD29A* showed that these genes were slightly up-regulated in the 35S: ZmDREB1Aa transgenic line. The above results showed that *ZmDREB1A* induces both ABA-independent genes like *COR15A, KIN1*, and *KIN2* and ABA-dependent genes like *RD17, ERD10*, and *RD29A*. Therefore, it was concluded that this gene might affect the expression of dehydration and cold-responsive genes in both the ABA-independent and ABA-dependent pathways. Likewise, another DREB TF gene *ZmDBP3* was highly induced by cold and moderately induced by salinity stress (*Wang & Dong, 2009*) (Table 1). Overexpression of this gene in transgenic *Arabidopsis* led to improved tolerance to both cold and drought stresses (*Chang & Yin, 2009*) (Table 2).

Natural variations present in the promoter region of another maize DREB TF gene *ZmDREB2*, lead to drought tolerance in maize (*Liu et al., 2013*) (Table 1). In transgenic *Arabidopsis*, overexpression of *ZmDREB2* resulted in plants with enhanced tolerance to drought. Elsewhere, qRT-PCR analysis of maize leaves revealed that expression of *ZmDREB2A* was induced by dehydration, heat and cold stress (*Qin et al., 2007*). Overexpression of *ZmDREB2A* in transgenic *Arabidopsis* resulted in dwarf plants with enhanced tolerance to drought and heat stresses. Microarray analysis of these transgenic

**Table 2 Abiotic stress responses of overexpressing Maize TFs in transgenic plants.**

| Family | Gene | Stress tolerance | Transgenic plant | References |
|--------|------|------------------|------------------|------------|
| MYB/MYC | ZmMYB30 | Salt | Arabidopsis | Chen et al. (2017) |
| | ZmMYB31 | Sensitivity to UV irradiation | Arabidopsis | Fornalé et al. (2010) |
| DREB/CBF | ZmDREB2A | Drought, heat | Arabidopsis | Qin et al. (2007) |
| | ZmDBP3 | Cold, salt | Arabidopsis | Wang & Dong (2009) and Chang & Yin (2009) |
| | ZmDBF3 | Salt, freezing | Yeast (Saccharomyces cerevisiae) | Zhou et al. (2016) |
| | ZmDREB1A | Drought, Freezing | Arabidopsis | Qin et al. (2004) |
| | ZmDREB2.7 | Drought | Arabidopsis | Liu et al. (2013) |
| | ZmDBP4 | Drought, cold | Arabidopsis | Wang, Yang & Yang (2011) |
| bZIP | ZmbZIP60 | Dithiothreitol (DDT) | Arabidopsis | Wang et al. (2012) |
| | ZmbZIP72 | Drought, partial salinity | Arabidopsis | Ying et al. (2012) |
| | Abp9 | Salt, osmotic stress | Cotton (Gossypium hirsutum) | Wang et al. (2017) |
| | | Drought, ABA, salt | Arabidopsis | Zhang et al. (2011) |
| NAC | ZmSNAC1 | Sensitivity to ABA, osmotic stress | Arabidopsis | Lu et al. (2012) |
| | | Tolerance to dehydration | | |
| | ZmNAC55 | Sensitivity to ABA | Arabidopsis | Mao et al. (2016) |
| | | Tolerance to drought | | |
| | ZmNAC111 | Drought | Maize (Zea mays) | Mao et al. (2015) |
| WRKY | ZmWRKY17 | Sensitivity to salt | Arabidopsis | Cai et al. (2017) |
| | | Tolerance to ABA | | |
| | ZmWRKY33 | Salt | Arabidopsis | Li et al. (2013) |
| | ZmWRKY44 | Sensitivity to salt | Arabidopsis | Wang (2014) |
| | ZmWRKY58 | Enhanced tolerance to drought, salt | Rice (Oryza sativa) | Cai et al. (2014) |
| | ZmWRKY106 | Drought, heat stress | Arabidopsis | Wang et al. (2018c) |
| | ZmWRKY40 | drought | Arabidopsis | Wang et al. (2018a) |
| Others | | | | |
| HSF | ZmHsf06 | Drought, thermotolerance | Arabidopsis | Li et al. (2015) |
| HD-Zip | Zmhdz10 | Drought, salt sensitivity to ABA | Rice (Oryza sativa) | Zhao et al. (2018) |
| | | Drought, salt | Arabidopsis | |
| | ZmHDZ4 | Drought | Rice (Oryza sativa) | Wu et al. (2016) |
| | Zmhdz12 | Drought | Arabidopsis | Qing & Wei (2018) |
| NF-Y | ZmNF-YB2 | Drought | Maize (Zea mays) | Nelson et al. (2007) |
| | ZmNF-YB16 | Drought, dehydration | Maize (Zea mays) | Wang et al. (2018b) |

**Note:**
Represents the abiotic stress responses of overexpressing Maize TFs in transgenic plants.

*Arabidopsis* plants identified a number of genes associated with detoxification and heat shock (HS) for example *RD29B* and *At5G03720*. Moreover, five genes coding for LEA proteins (*LEA14, At1g52690, At3G53040, At3G15670,* and *At2G36640*) in addition to a metabolism associated gene *AtGoIS3*, were all up-regulated under different stress treatments in the transgenic lines (Table 1). Elsewhere, functional analysis of *ZmDBF3* showed that this TF gene was activated by drought, high temperature, salt, cold, and ABA. However, no significant difference was noted under methyl jasmonate and SA (*Zhou et al., 2016*). Ectopic expression

of *ZmDBF3* in yeast (*Saccharomyces cerevisiae*) resulted in a higher survival rate during exposure to KCI, $Na_2CO_3$, $NaHCO_3$, NaCl, PEG 6000, sorbitol, and freezing temperatures. Moreover, exogenous expression of *ZmDBF3* in transgenic *Arabidopsis* considerably improved tolerance to drought, freezing, and salt stresses (Table 2). These findings, suggest that *ZmDBF3*, a novel maize DREB TF may have similar functions to a regulatory factor taking part in abiotic stress response pathways. Similarly, overexpression of *ZmDBP4* in *Arabidopsis* resulted in transgenic plants with improved cold and drought stress tolerance (*Wang, Yang & Yang, 2011*) (Table 2). Analysis of the promoter region of *ZmDBP4* identified *cis*-acting elements which responded to abiotic stresses, suggesting that *ZmDBP4* encodes a functional factor that plays an important role in the control of multiple abiotic stress responses in maize. Elsewhere, mRNA accumulation analysis profiles of two DRE-binding proteins (*DBF1* and *DBF2*) in maize seedlings revealed that *DBF1* was induced during embryogenesis and in response to drought, ABA, and NaCl treatments (*Kizis & Pages, 2002*).

Dehydration responsive element binding TFs are versatile when it comes to abiotic stress regulation. Recently, numerous studies have been done to understand the roles DREB TFs play in abiotic stress responses and to reveal the mechanisms involved in their transcription and post transcriptional regulation. Collectively, these studies suggest that DREB TFs can be potential candidates for abiotic stress tolerance, although these studies have not addressed the vital question of whether DREB TFs can improve the yield of engineered crops under stress. Many DREB homologues have been identified and isolated from different plant species, especially from plant species with exceptional tolerance to different abiotic stresses. Thus, the focus now is to evaluate the existing methods of yield analyses under different stress conditions and to assess transgenic plants in actual field conditions.

### NAC TFs and regulons

The TF members in the NAC family, (ATAF, CUC, and NAM) represent one of the largest plant-specific TFs (*Ooka et al., 2003*). In the main crop species, a large number of NAC TFs have been analyzed and sequenced at the genome-wide level. These include 151 members in rice and 117 in *Arabidopsis* (*Nuruzzaman et al., 2010*), 204 members in the Chinese cabbage (*Liu et al., 2014b*), and 152 members in maize (*Shiriga et al., 2014*). The TFs belonging to the NAC family share a greatly conserved N-terminus made up of 150–160 amino acid residues, constituting a DNA-DB that carries five sub-domains (A–E) and a varying C-terminal (*Hu et al., 2008*; *Ooka et al., 2003*). The NAC genes and their constituent *cis*-acting elements make up the NAC regulons, which further provide vital examples of finely characterized collaboration between a single TF and one or more *cis*-acting elements that associate in response to multiple stresses (*Christianson et al., 2010*). The roles of NAC TFs in plants have been extensively studied in rice and *Arabidopsis*. In *Arabidopsis*, for example, an *ERD1* (early dehydration stress 1) gene was activated by a number of NAC TFs including *ANAC055*, *ANAC019*, and *ANAC072* (*Tran et al., 2007*). A rice NAC TF designated as OsNAM, was found to regulate the activation of five genes (*OsAH*, *OsCESA*, *OsMtN3*, *OsGdpD*, and *OsGDP*) in response to drought (*Dixit et al., 2015*). Several NAC TFs utilize the NACRS motif in plants, for

instance *SNAC2* and *ENAC1* found in rice (Sun et al., 2012) and *ANAC055*, *ANAC072*, and *ANAC019* found in *Arabidopsis* (Tran et al., 2004).

In maize, several NAC TFs involved in abiotic stress regulatory pathways have been isolated, cloned, and characterized. Recently, expression analysis of *ZmSNAC1* in maize seedlings revealed that this TF gene was strongly induced by high salinity, drought, ABA treatment, and low temperature, although it was down-regulated in response to SA treatment (Lu et al., 2012). Overexpression of *ZmSNAC1* in transgenic *Arabidopsis* led to increased hypersensitivity to osmotic stress and ABA as well as enhanced tolerance to dehydration stress at the germination phase (Table 2). These results suggest that *ZmSNAC1* acts as a multiple stress responsive TF, positively modulating abiotic stress tolerance in maize. Elsewhere, Shiriga et al. (2014) identified 11 NAC TF genes in maize that were induced by various abiotic stresses. This prediction was confirmed when these genes were differentially expressed in response to drought stress. Four genes, *ZmNAC45*, *ZmNAC72*, *ZmNAC18*, and *ZmNAC51* were all up regulated in the drought-tolerant maize genotypes and down-regulated in the drought susceptible genotypes. Recently, seven ZmNTL, NAC TFs genes (*ZmNTL1, ZmNTL2, ZmNTL3, ZmNTL4, ZmNTL5, ZmNTL6*, and *ZmNTL7*) were analyzed in maize seedlings and all seven genes were found to be strongly expressed in the stem and roots and down-regulated in the leaves when the plants were exposed to $H_2O_2$ and/or ABA treatments. Exogenous expression of *ZmNTL1, ZmNTL2*, and *ZmNTL5* in transgenic *Arabidopsis* led to increased tolerance to $H_2O_2$ in transgenic plants (Wang et al., 2016a). Overexpression of *ZmNAC55* in transgenic *Arabidopsis* resulted in plants which were hypersensitive to ABA at the seedling stage but showed enhanced resistance to drought when compared to the wild-type control seedlings (Mao et al., 2016). Additionally, 12 stress-responsive genes (*RD20, NCED3, ZAT10, ANAC019, LEA14, RD29B, RD29A, DREB2A, RD17, RD26, RAB18*, and *PP2CA*) were all up regulated in response to drought stress in the transgenic lines (Table 1). Expression profiles of *ZmNAC55* in maize revealed that this gene was induced by high salinity, drought, ABA, and cold stress.

Elsewhere, seven NAC TF genes analyzed in maize seedlings (*Zma001259, Zma000584, Zma029150, ZmSNAC052, Zma003086, Zma054594*, and *Zma006493*) were all found to be up regulated in response to salt stress in all tissues (Lu et al., 2015). In response to PEG treatment, three of the above-mentioned genes, namely *Zma006493, Zma003086*, and *Zma000584* were significantly up regulated in the roots only, while *Zma001259, Zma029150, Zma000584*, and *Zma054594* were all strongly expressed in both the roots and shoots. Five genes, *Zma054594, Zma000584, Zma001259, Zma003086*, and *ZmSNAC052* were activated by cold stress although in varying degrees. In conclusion, due to the strong expression in response to ABA treatments, these seven genes could play a vital role in the ABA-dependent signaling network in maize.

Numerous advancements in NAC TFs functional studies have been achieved over the past few years. However, most of these studies are related to the involvement of NAC TFs in biotic stress responses. To achieve a deeper understanding of NAC TFs in abiotic stress responses, it is of vital importance to identify the main components of signal transduction pathways that interact with these TFs. Utilizing data obtained from microarray

analyses could help in the direct determination of specific NAC DNA-binding sites on a global scale under different abiotic stress conditions.

Finally, numerous studies have demonstrated the use of stress-responsive NAC TFs in the improvement of abiotic stress tolerance in crops by genetic engineering. In view of the specificity of NAC TF in multiple stress responses, NAC TFs that are induced by multiple abiotic stresses are promising candidates in the engineering of plant varieties with improved multiple stress tolerance (*Shao, Wang & Tang, 2015*). Moreover, field evaluation of engineered crops containing NAC TF genes and efficient promoters, for reducing detrimental effects triggered by overexpression of some NAC genes must be considered (*Rushton et al., 2008*).

### bZIP TFs: AREB/ABF regulon

The ABRE (PyACGTGG/TC), is a conserved *cis*-acting element bound by the basic Leucine Zipper Domain (bZIP) TFs (*Ciarmiello et al., 2011*). The ABRE was first established on the promoter region of ABA-activated genes by *Giraudat et al. (1994)*. The bZIP TFs belong to one of the largest and diversified TF families in plants. They are categorized into ten subfamilies based on the presence of extra conserved motifs and the basic region sequence similarities (*Pérez-Rodríguez et al., 2010*). AREB/ABF TFs are characterized by a strongly conserved bZIP domain made up of two structural components (a leucine (Leu) zipper and a basic region), the Leu zipper is composed of heptad repeats of Leu and/or other heavy hydrophobic amino acid residues and controls hetero- and or homodimerization of the bZIP proteins. The basic region is composed of 16 amino acids with the indistinguishable N-x7-R/K-x9 motif and is responsible for DNA binding and nuclear localization (*Jakoby et al., 2002*). The bZIP TFs, which are part of the AREB/ABF regulons, give an excellent example of interactions involving stress-responsive genes and TFs carrying the *cis*-acting element (ABRE). In maize, a bZIP TF gene *ABP9* that has the ability to bind to the AREB2 motif located in the *Cat1* promoter region was activated by drought, salt, $H_2O_2$, and ABA (*Zhang et al., 2011*). Exogenous expression of *ABP9* in *Arabidopsis* led to significant tolerance to freezing, salt, oxidative stress, and drought in transgenic plants. Transgenic *Arabidopsis* plants also showed enhanced sensitivity to exogenously supplied ABA during stomatal closure, seed germination, and root growth. Furthermore, transgenic plants expressing *ABP9* showed reduced levels of oxidative cellular damage, reduced cell death, and reduced levels of ROS.

More recently, *Wang et al. (2017)* demonstrated that *ABP9* enhanced salt and osmotic stress tolerance in transgenic cotton plants. Overexpression of *ABP9* resulted in elevated transcripts of several stress responsive-genes (*GhNCED2, GhDBP2, GhZFP1, GhHB1, GhSAP1*, and *GhERF1*) in the transgenic cotton plants in response to salt stress (Table 2). Additionally, transgenic plants were shown to have higher germination rates, and improved root systems in a greenhouse setting and reduced stomatal density and stomatal aperture in a growth room. Finally, the relative water content and survival rate of the transgenic plants was significantly higher compared to the control plants in response to drought. *Wang et al. (2012)* demonstrated that expression of *ZmbZIP60* was highly activated by a wide range of stresses including ABA, high salinity, tunicamycin treatment,

and dehydration (Table 1). In the wild-type *Arabidopsis*, overexpression of *ZmbZIP60* resulted in plants with enhanced tolerance to dithiothreitol stress. Furthermore, *Li et al. (2018)* discovered a major QTL governing heat-induced *ZmbZIP60* expression and deduced that the upstream region of *ZmbZIP60* plays a vital role in regulating responses to heat stress in maize.

Similarly, *Ying et al. (2012)* cloned and characterized a maize bZIP TF gene designated as *ZmbZIP72*, which was induced by drought, ABA, and high salinity stress (Table 1). *ZmbZIP72* was differentially expressed in various organs in maize. Overexpression of *ZmbZIP72* in transgenic *Arabidopsis* led to enhanced tolerance to drought, partial tolerance to salinity and hypersensitivity to osmotic stress and ABA treatment. Furthermore, the transgenic *Arabidopsis* plants also showed enhanced expression of several ABA-inducible genes including (*RAB18*, *HIS1-3*, and *RD29B*). Elsewhere, microarray analysis of two specific maize inbred lines, a drought-sensitive Ye478 line, and a drought tolerant Han21 line revealed that 22 ZmbZIP genes might play a critical role in drought tolerance (*Wei et al., 2012a*). In the same report, *ZmbZIP37* an orthologous gene of two rice genes *OsbZIP72* and *OsbZIP23* that both play vital roles in drought tolerance and ABA response in rice, was found to be up-regulated in response to drought stress in maize. Similarly, cloning and characterization of a bZIP TF gene *ZmbZIP17* from the Han21 maize inbred line revealed that this gene was up regulated in response to drought (*Jia et al., 2009*). Real-time PCR analysis revealed that *ZmbZIP17* was highly up regulated in response to heat, salinity, drought, and ABA stresses immediately, suggesting that this gene represents an early responsive gene that reacts to various abiotic stresses. Elsewhere, expression analysis of two maize bZIP TF genes *ZmbZIP107* and *ZmbZIP54* revealed that these two genes were highly elevated in a lead tolerant maize line when compared to a lead sensitive line in response to different treatments of lead (*Zhang et al., 2017a*) (Table 1). Recently, *Ma et al. (2018)* demonstrated that *ZmbZIP4* was induced by drought, cold, high salinity, ABA, and heat in maize seedlings. Overexpression of *ZmbZIP4* led to an improved root system, increase in the number of lateral roots, and longer primary roots in transgenic maize. Additionally, genome-wide analysis of *ZmbZIP4* target genes by immunoprecipitation sequencing, unearthed a number of downstream stress response genes that were positively regulated by *ZmbZIP4*. These downstream target genes included *ZmRD21, ZmLEA2, ZmRD20, ZmGEA6, ZmNHX3*, and *ZmRAB18*. Collectively, these results suggested that *ZmbZIP4* is a positive regulator of abiotic stress response that takes part in root development in maize.

In conclusion, the promoter region of each abiotic stress responsive gene might carry a single or several proximal or distal coupling elements (CE) for instance, CE 3 and CE1 which activate the expression of abiotic stress-responsive genes. In addition, *Shen, Zhang & Ho (1996)* identified CEs in *Hordeum vulgare* that form an abscisic acid response complex, which could be a necessary component in triggering ABA-mediated gene expression. Collectively, the above reports confirmed the participation of bZIP TFs in the ABA signaling pathway. These findings could be useful in the future development of better genotypes with improved tolerance to various abiotic stresses (*Todaka, Shinozaki & Yamaguchi-Shinozaki, 2015*). An accurate understanding of the functions of bZIP TFs in

crops will require an accurate mapping of the location of bZIP genes in the different plant organs.

### WRKY TFs and WRKY regulons

WRKY proteins represent the largest superfamily of TFs, which are specific to plants. WRKY TFs control plant growth and development and spur tolerance against both abiotic and biotic stresses (*Tripathi, Rabara & Rushton, 2014*). WRKY TFs are usually identified by a WRKY domain made up of 60 amino acid residues, and contains a highly conserved WRKYGQK sequence followed-up by a zinc-finger motif. The WRKY domain shows a strong binding affinity for a *cis*-acting element known as W-box (TTGACC/T), which is present in a number of abiotic stress responsive genes (*Rushton et al., 2010*; *Ulker & Somssich, 2004*).

Several WRKY TFs involved in abiotic stress tolerance have recently been reported in maize. For example, functional analysis of *ZmWRKY33* under different abiotic stresses, revealed that this gene was activated by cold, dehydration, ABA, and salt treatments (*Li et al., 2013*). Overexpression of *ZmWRKY33* in transgenic *Arabidopsis* led to the activation of two stress-activated genes (*RD29A* and *DREB1B*), which were both up-regulated resulting in enhanced salt tolerance in the transgenic plants (Table 1). The above-mentioned results strongly suggest that this maize WRKY TF plays a vital role in abiotic stress regulation in maize. Elsewhere, *Wang (2014)* demonstrated that exogenous overexpression of *ZmWRKY44* in transgenic *Arabidopsis* resulted in plants that were moderately sensitive to NaCl stress. In maize seedlings, ZmWRKY44 was induced by high temperature, salt stress, ABA, and $H_2O_2$ treatments. Recently, *ZmWRKY17* was cloned, characterized and its expression was analyzed in maize seedlings (*Cai et al., 2017*) (Table 1). The results showed *ZmWRKY17* was induced by ABA, salt, and drought stresses. Additionally, constitutive expression of this gene in transgenic *Arabidopsis* led to a striking reduction in tolerance to salt stress, as confirmed by the physiological assays performed on relative electrical leakage, MDA content, cotyledons greening rate and root growth. Still in the same study, RNA-Seq analysis showed that eight stress-related genes (*DREB1F, KIN1, bHLH92, RD29A, RD29B, NAC019, RD22*, and *MYB101*) were significantly up-regulated in the wild-type plants when compared to the transgenic plant lines in response to salt stress. However, expression of *NCED5* was higher in transgenic plants under the same stress. Together, these results give a strong indication that *ZmWRKY17* may function as a negative regulator in response to drought stress in maize. This could be due to elevated levels of ABA ensuing as a direct response to salt stress through the ABA signaling system. *Wei et al. (2012b)* compared the expression profiles of 31 WRKY genes in two maize lines, a drought-sensitive Ye478 line and a drought tolerant Han21 line. The results showed that the expression of the WRKY genes in the drought-tolerant Han21 line changed less, and the seedlings recovered faster when re-watered, as opposed to the drought-sensitive Ye478 seedlings. In the same study, the expression of *ZmWRKY115* was decreased as a direct result of drought stress. Elsewhere, qRT-PCR expression analysis showed that *ZmWRKY58* was activated by salt, drought and ABA treatments (*Cai et al., 2014*) (Table 1). Constitutive expression of *ZmWRKY58* in transgenic rice led to delayed germination and

constrained post-germination growth and development. However, transgenic seedlings overexpressing *ZmWRKY58* reported increased tolerance to both salt and drought stresses (Table 2). Similarly, *Wang et al. (2018a)* identified a WRKY TF gene named *ZmWRKY40* (Table 1). A number of stress-related transcriptional regulatory factors were located in the promoter region of this gene. In maize, *ZmWRKY40* was induced by high salinity, drought, ABA, and high temperature. Overexpression of *ZmWRKY40* in *Arabidopsis* led to enhanced drought tolerance in the transgenic plants. Additionally, overexpression of *ZmWRKY40* induced the expression of three stress-responsive genes *DREB2A*, *STZ*, and *RD29A* in transgenic *Arabidopsis*. Recently, the expression of *ZmWRKY106*, a member of the WRKYII group was found to be induced by high temperature, drought, and exogenous ABA treatment, but was weakly induced by salinity (*Wang et al., 2018c*). Overexpression of *ZmWRKY106* in transgenic Arabidopsis led to improved tolerance to heat and drought. Additionally, *ZmWRKY106* positively regulated the expression of several stress response genes including *RD29A*, *CuZnSOD*, *DREB2A*, and *NCED6*. The above results strongly indicate *ZmWRKY106* may play an important role in the abiotic stress response pathways in maize by regulating stress-related genes.

In the model plant *Arabidopsis*, two WRKY genes WRKY 60 and WRKY 18 were found to regulate ABA signaling positively while one WRKY gene *WRKY40* negatively regulated ABA signaling. These three WRKY genes mentioned above, bind to the promoter region of several genes including some TFs genes like (*DREB1A/CBF3, ABI5*, and *DREB2A*), and several stress-regulated genes like (*COR47* and *RD29A*) in the process controlling their expression (*Shang et al., 2010*; *Chen et al., 2010*).

WRKY TFs have been identified as promising candidates for crop improvement due to the strict regulations involved in the identification and binding of these TFs to the downstream target promoter regions (*Phukan, Jeena & Shukla, 2016*). Taken together, all the above insights highlight the multiple stress responses and diverse regulation of WRKY TFs in maize and other crops.

### Other TFs and their regulons

Apart from the five main TF families described above, other TF families take part in diverse roles in plants including, regulating responses to both abiotic and biotic stresses, and various growth and development processes. Recently, extensive research has uncovered stress-mitigating roles of a number of TFs whose responses to abiotic stressors were previously unknown in maize. Three of these TF families are briefly described below.

### Homeodomain-leucine zipper I

Homeodomain-leucine zipper (HD-Zip) proteins represent a large TF family that is specific to plants. HD-Zip proteins have been cloned and characterized in several important crops and some model plants such as rice, *Arabidopsis*, tomato, and sunflower (*Johannesson et al., 2003*; *Lin et al., 2008*; *Agalou et al., 2008*; *Manavella et al., 2006*). HD-Zip proteins are characterized by a DNA-binding HD and a neighboring leucine zipper (Zip) motif whose function is to mediate protein dimerization (*Ariel et al., 2007*). HD-Zip proteins belonging to the subfamily I are believed to take part in the majority of

plant responses to abiotic stresses (*Ariel et al., 2007*). In *Arabidopsis* for example, analysis of four HD-Zip TFs (*ATHB6*, *ATHB7*, *ATHB5*, and *ATHB12*) revealed that these genes were up-regulated or repressed by either ABA or drought stress (*Soderman, Mattsson & Engstrom, 1996*; *Soderman et al., 1999*; *Lee et al., 2001*; *Johannesson et al., 2003*). These results suggest that these four genes may play a vital role in the regulation of abiotic stress regulatory networks in plants.

In maize, *Zmhdz10* was the first HD-Zip TF to be isolated and characterized (*Zhao et al., 2014*). Expression of this gene was activated by ABA treatment and salinity stress (Table 1). Exogenous overexpression of *Zmhdz10* in transgenic rice resulted in improved tolerance to salt and drought stress and enhanced sensitivity to ABA. Furthermore, the transgenic plants had elevated levels of proline and reduced MDA content when compared to the wild-type plants (Table 2). Transgenic *Arabidopsis* plants overexpressing *Zmhdz10* exhibited strong tolerance to salt and drought stresses, at the same time, expression patterns of several ABA-responsive genes namely (*ABI1, RD29B, P5CS1,* and *RD22*) were altered. The above results give a strong indication that *Zmhdz10* serves as a transcriptional regulator that can positively regulate both salt and drought stress tolerance in the ABA-dependent pathway in plants. Recently, *Qing & Wei (2018)* isolated and characterized a maize HD-ZIP TF designated as *Zmhdz12*. Tissue expression analysis revealed that this TF was strongly expressed in the leaves compared to other tissues. In transgenic *Arabidopsis*, *Zmhdz12* was activated by drought as observed when the drought resistant transgenic lines were compared to the wild-type lines. Similarly, expression status of 17 *Zmhdz* I genes from maize (*Zmhdz1* to *Zmhdz17*) revealed that all these genes were either repressed or up-regulated due to drought stress (*Zhao et al., 2011*). Additionally, many of the genes above belonging to the same subgroup in the phylogenetic tree, showed similar patterns of expressions. Elsewhere, *ZmHDZ4* was isolated and characterized in maize for its role in drought stress tolerance (*Wu et al., 2016*). Overexpression of *ZmHDZ4* in transgenic rice resulted in plants with enhanced tolerance to drought.

In conclusion, it is worth noting that HD-Zip proteins play crucial roles in cuticle formation, so they might be involved in abiotic stress tolerance and protection against plant pathogens (*Chew, Hrmova & Lopato, 2013*). In addition, the roles described above make HD-Zip TFs ideal candidates for genetic engineering in maize and other major crops. Although more in-depth studies are needed in order to ascertain the function of individual HD-Zip family members in response to various abiotic stresses.

## Heat shock proteins

All organisms are composed of an evolutionarily conserved, fast cellular defense system known as HS response, which regulates various reactions associated with heat stress and a variety of chemical stressors (*Lin et al., 2011*). Heat shock proteins (HSPs) were first discovered in the salivary glands of *Drosophila* in response to HS (*Ashburner & Bonner, 1979*). HSFs family members function by binding to the promoter of chaperones referred to as HSPs. HSF TFs have a three N-terminal section and a C-terminal section in addition to leucine amino acid (*Schuetz et al., 1991*). HSPs can be categorized into six main

families, (Hsp90, Hsp40, Hsp90, Hsp60, Hsp70, and Hsp110) based on their molecular sizes (*Wang et al., 2004*). HSPs in plants were first characterized in tomato (*Scharf et al., 1990*), and since then more HSFs have been reported in other plants such as *Arabidopsis*, rice, sunflower, and wheat (*Hübel & Schöffl, 1994*; *Yamanouchi et al., 2002*; *Almoguera et al., 2002*; *Shim et al., 2009*). A survey recently reported that there are at least 24 HSFs in *Brachypodium*, 21 in *Arabidopsis*, 30 in maize, 25 in rice, 52 in soybeans, and 27 in tomatoes (*Scharf et al., 2012*). In the model plant *Arabidopsis*, a HsfA2 mutant displayed tolerance to osmotic stress, salt, and heat stresses, suggesting that this gene is involved in several abiotic stress response networks and pathways (*Ogawa, Yamaguchi & Nishiuchi, 2007*).

Few HSPs gene have been isolated and characterized in maize. *Song et al. (2016)* isolated and characterized a Hsp70 gene named *ZmERD2* (Early Responsive to Dehydration 2) from maize (Table 1). Expression patterns of *ZmERD2* revealed that this gene was induced by cold, high salinity, dehydration, heat stress, and PEG but was not induced by ABA. Further expression analysis revealed *ZmERD2* was instantly activated at 42 °C and its peak was reached after 1 h of exposure to heat stress. Elsewhere, expression patterns of 22 Hsf genes from maize showed that these genes were differentially expressed when subjected to heat stress (*Lin et al., 2011*). Further analyses from this study revealed that *ZmHsfA2* subclass in maize has close relations with HS response. This is after three HsfA2 genes (*ZmHsf-17*, *ZmHsf-01*, and *ZmHsf-04*) were strongly expressed in response to heat stress. In addition, six more genes were highly up-regulated in response to heat stress (*ZmHsf-03*, *ZmHsf-01*, *ZmHsf-23*, *ZmHsf-24*, *ZmHsf-04*, and *ZmHsf-25*). These results pointing to the specific roles these genes play in maize in response to heat stress. *Li et al. (2015)* recently cloned a maize HSF designated as *ZmHsf06* from maize and transformed it in *Arabidopsis thaliana*. Expression analysis of the transgenic plants overexpressing *ZmHsf06* revealed that this gene was induced by drought and heat stress (Table 2). The above results were confirmed by biochemical and physiological evidence that showed that the transgenic plants displayed longer axial root length, higher seed germination rate, elevated levels of chlorophyll in leaves as well as reduced osmotic potential and MDA content when compared to the wild-type plants. Based on the above results, it's evident *ZmHsf06* could have future potential use in molecular breeding in maize as well as other crops for improved drought and heat stress tolerance.

Collectively, it is important to mention that HSPs have been shown to have a close association with ROS, meaning that plants have gained a stronger level of ROS regulation throughout the course of evolution (*Banti et al., 2010*). Therefore, understanding the roles played by HSPs in plant responses to abiotic stresses will be useful in the engineering of abiotic stress tolerant crop varieties. HSP have been studied and characterized in a number of important crop varieties as mentioned above, although their functional plasticity, and genome sequence data is still limited (*Echevarría-Zomeño et al., 2016*).

## NF-Y transcription factors

Nuclear factor Y also referred to as CBF (CCAAT binding factor) or heme activator protein (HAP), is a complex made up of three subunits NF-YB (CBF-A or HAP3), NF-YA

(CBF-B or HAP2) and NF-YC (CBF-C or HAP5) (*Nardini et al., 2013*; *Wang et al., 2018b*). The NF-Y TF family has been comprehensively studied in animal systems, and it was found that each subunit is encoded by a single gene in yeast and mammals (*Mantovani, 1999*). NF-Y TFs interact with other factors in the regulatory network to induce or inhibit the expression of downstream target genes (*Benatti et al., 2008*). Unlike mammals and yeast, plants have many NF-Y subunit genes (*Wang et al., 2018b*). For example, in *Arabidopsis* 13 genes encoding NF-YB, 10 genes encoding NF-YA, and 13 genes encoding NF-YC have been reported (*Siefers et al., 2009*). Additionally, individual NF-Y subunits have been shown to play vital roles in plant abiotic stress tolerance (*Sato et al., 2014*; *Ma et al., 2015*).

Even though maize has numerous NF-Y subunits, very few studies have been done to investigate the roles these subunits play in response to abiotic stress (*Wang et al., 2018b*). *Nelson et al. (2007)* demonstrated that transgenic maize with elevated levels of *ZmNF-YB2* showed improved tolerance to drought stress based on responses from various stress-related parameters which included stomatal conductance, chlorophyll content, reduced wilting, and leaf temperature (Table 2). Recently, overexpression of a NF-YB TF complex member designated as *ZmNF-YB16* resulted in improved drought and dehydration resistance in transgenic inbred maize line B104 during reproductive and vegetative stages (*Wang et al., 2018b*) (Table 1). Analysis of gene expression in the photosynthesis system between the WT and transgenic plants revealed that several genes were up-regulated in the transgenic plants when compared to the WT plants. Examples of genes up-regulated included GRMZM2G117572 (encoding the photosystem II PsbZ protein), GRMZM2G414660 (encoding the photosystem II cytochrome b599 subunit) and GRMZM5G831399 (encoding the photosystem II PsbH protein) among others.

Analysis of the co-expression between miR169, miRNA family, and ZmNF-YA TFs in transgenic *Nicotiana bethamiana* revealed that mutations in deletion sites terminate the regulation of zma-miR169 (*Luan et al., 2014*). The expression levels of *zma-miR169l*, *zma-miR169i*, and *zma-miR169a* were all inversely correlated with *ZmNF-YA11*, *ZmNF-YA6*, and *ZmNF-YA7* over the short term. However, over the long term, the expression levels of all the NF-YA genes and miR169s decreased, revealing that *ZmNF-YA11, ZmNF-YA6*, and *ZmNF-YA7* could not have been regulated by zma-miR169 in response to PEG stress after 15 days. Majority of the zma-miR169s were up-regulated by external ABA and down-regulated by drought stress but showed an early increase in expression and later a decline in response to salinity stress. Recently, *Su et al. (2018)* identified a NF-Y TF designated as *ZmNF-YA3*. Genome-wide analysis revealed that *ZmNF-YA3* was linked to more than 6,000 sites in the maize genome, 2,259 of which are linked with genic sequences. Moreover, it was shown that *ZmNF-YA3* could significantly improve high temperature and drought tolerance in maize by binding to the promoter region of three downstream genes (*ZmMYC4, ZmbHLH92*, and *ZmFAMA*).

In conclusion, all of the insights obtained above suggest NF-Y TFs play an important role in abiotic stress tolerance in maize by regulating several vital downstream genes involved in important aspects of abiotic stress responses, and plant growth and development, for instance, photosynthesis and ER stress response. Therefore, NF-Y TF

genes could be engineered in maize and other crops in order to improve their abiotic stress tolerance, leading to improved production.

### Engineering of TFs

The recent discovery of TFs as potential tools in the manipulation and engineering of quantitative traits such as drought and salinity has ignited the development of novel technologies based on TFs and benefiting not only gene discovery but also crop improvement. Engineering of TF activity has been a major target in these efforts, a direction that offers future promises in modulating metabolic pathways. For example, overexpression of DREB2 resulted in no stress tolerance improvement because proteins are composed of domains that limit the induction of their target genes downstream (*Liu et al., 1998*). *Sakuma et al. (2006)* obtained drought-tolerant plants by removing this repressor function through the engineering of point mutations. An undesirable effect of overexpressing stress-related TFs is that sometimes it negatively affects the growth and development of a plant leading to stunted growth or toxicity (*Hussain, Amjad & Amjad, 2011*). Elsewhere, overexpression of *ZmDREB2A* under a stress-activated promoter in the transgenic plants led to significant improvement in drought tolerance (*Qin et al., 2007*).

Transcriptional down regulators that repress gene expression in response to various abiotic stresses are also important tools in manipulating drought tolerance. For example, overexpression of a yeast transcription repressor *CaZPF1* in *Arabidopsis* led to drought tolerance in transgenic plants (*Kim et al., 2004*). In the model plant *Arabidopsis*, systematic analysis of TF families resulted in the discovery of target genes that have the potential to enhance abiotic stress tolerance in major crops (*Riechmann et al., 2000*). A good example is the discovery of *AtNF-YB1* gene that belongs to the Nuclear factor Y TF family (*Nelson et al., 2007*). The orthologue of (*AtNF-YB1*) gene in maize (*ZmNF-YB2*), when overexpressed in transgenic maize, resulted in drought-tolerant crops (Table 2). These findings emphasized the significance of TFs, especially when used in the engineering of plants.

The abiotic stress response networks in plants are very complex due to the large number of gene families involved and the complicated associations between the *cis*-acting elements and the TFs. In addition, a single TF may regulate a large number of target genes with similar *cis*-elements whereas TFs from different families may regulate a single target gene with different types of *cis*-acting elements (*Hussain, Amjad & Amjad, 2011*). Therefore, abiotic stresses regulating TFs not only function independently but also co-regulate abiotic stress responses between each other (*Wang, Shao & Tang, 2016b*) (Fig. 2). As mentioned throughout this review, genetic engineering of TF genes will be much more significant than manipulating a single functional gene. Thereafter, validation of the identified genes should be performed in model crops as well as the main crops by utilizing a stress-inducible promoter to reduce the detrimental effects brought about because of overexpression of certain TFs (*Hoang et al., 2017*). Moving forward, all of these advances will help elucidate the detailed regulatory channels taking part in multiple abiotic stress responses in plants, leading to the acquisition of target TF genes for enhanced breeding of abiotic stress-tolerant plants with improved desirable qualities and yields.

## Current and post genomics approaches

Abiotic stresses represent a combination of various distinct traits consisting of a quantitative pattern of inheritance. Therefore, in order to efficiently understand the plants response to the different abiotic stresses at the molecular level, a deeper understanding of the systems involved in transcription regulation is required. Trait mapping, functional characterization, genomic selection, rapid RNA, and DNA high-throughput SNP genotyping tools, sequencing technologies, and other platforms are currently used to analyze the genetic mechanisms of different abiotic stresses including drought, salinity, and cold in an effort to speed up the breeding process in maize (*Nepolean et al., 2018*).

Genome editing techniques are the most recent technologies applied in gene function analyses and manipulations. RNA interference is a rapid and inexpensive technique utilized to analyze gene function in targeted gene knockdown analyses (*Rabara, Tripathi & Rushton, 2014*). However, a disadvantage of this technique is that the inhibition of gene function is not complete and this could lead to unintended off-target effects leading to misinterpretation of results (*Gaj, Gersbach & Barbas, 2013*). Of the targeted genome editing approaches currently available, clustered regulatory interspaced short palindromic repeats "CRISPR" is the most effective system used in editing plant genomes (*Cong et al., 2013*). CRISPR artificial transcription factors (CRISPR-ATFs) are gaining popularity as an effective system for regulating in vivo plant gene expression (*Lowder et al., 2018*). For instance, two novel systems (CRISPR-Act2.0 and mTALE-Act) were developed that could be used to study GRNs and the control mechanisms involved in plants (*Lowder et al., 2018*).

The other systems are transcription activator-like effectors nucleases (*Boch et al., 2009*) and Zinc-finger nuclease (*Kim, Cha & Chandrasegaran, 1996*). Another approach with huge potentials in functional genomics in plants is targeting-induced local lesions in genomes (TILLING). For example, a TILLING approach known as ecotilling, which was used to identify variations in natural populations, was successfully used to identify TFs in rice associated with drought tolerance (*Yu et al., 2012*). These techniques are vital in the selection of better quality genotypes and target genes in the abiotic stress tolerance research in maize.

Another approach currently being explored is the use of machine learning in the study of TFs GRN. Predictions of TFBSs and their corresponding transcription factor target genes (TFTGs) using machine learning approaches has made substantial contributions to the study of GRNs (*Cui et al., 2014*). Understanding the interplay between TFs, TFBSs, and TFTGs is vital in understanding the mechanisms involved in the gene regulatory processes taking place during biotic and abiotic stress responses in plant (*Shinozaki & Yamaguchi-Shinozaki, 2006*). Various computational algorithms are available in form of software packages. Additionally, expansive use of these software packages has revealed that even though some techniques were developed for one species, the same techniques can be used to analyze the dataset from other species (*Cui et al., 2014*). For example, a combination of Context Likelihood of Relatedness algorithm analyzed on *Escherichia coli* (*Faith et al., 2007*), Double two-way *t*-tests algorithms analyzed on *Escherichia coli* and Learning Module

Networks algorithm tested on yeast (*Joshi et al., 2009*) was used to identify the presence of oxidative stress regulatory TFs in Arabidopsis (*Vermeirssen et al., 2014*). Moreover, The algorithm for the reconstruction of GRNs (*Margolin et al., 2006*) was established to deduce transcriptional regulations in human immune B cells, but was later used to deduce transcriptional interactions regulating root physiological and developmental processes in Arabidopsis (*Chávez Montes et al., 2014*).

Gene regulatory networks provide insights into the relationships between TFs and their corresponding target genes (*Koryachko et al., 2015*). For instance, network component analysis (NCA), a computational method developed for analysis of TF-gene interactions in microbial TF-GRNs, was employed to quantitatively analyze TF-GRNs critical in floral development in Arabidopsis (*Misra & Sriram, 2013*; *Ni et al., 2016*). The results showed that the NCA model adequately accounted for the total gene expression analysis in a TF-GRN of seven TFs (AG, HY5, SEPALLATA3, AP3/P1, AG, AP2, and AGL15) and 55 genes. Strong interactions were present between different TF-gene pairs, such as, *LFY* and *MYB17*, *AG* and *CRC*, *AP2* and *RD20*, *AGL15* and *RAV2*, and finally *HY5* and *HLH1*. In maize, a machine learning algorithm GENIE3 was used together with numerous RNA-Seq expression data to establish a four tissue (root, SAM, seed, and leaf) specific GRNs (*Huang et al., 2018*). The results showed that even though many TFs were expressed in multiple tissues, a multi-level examination predicted regulatory roles for many TFs. Additionally, 76.6% (30,028/39,479) of the genes were found to be expressed in all the maize tissues. Out of the total of 2,587 TFs annotated by GRASSIUS in maize (*Chen et al., 2013*), 54.46% were expressed in all the four tissues while 86.63% of the total TFs were expressed in at least one of the four tissues.

Understanding the mechanisms of GRNs is vital in gaining insights on how TFs control gene expression in response to various abiotic stresses. Wet lab experiments are technically demanding, time-consuming and financially demanding (*Penfold & Wild, 2011*). Many machine-learning approaches have been proposed with an aim of reducing costs and time spent in the prediction of GRNs. For instance, recent development of a publicly accessible maize TF ORF collection (TFome) consisting of 2,034 clones that correspond to 2,017 unique maize TFs and co-regulators (CoREGs), has vastly contributed to the better understanding of GRNs (*Burdo et al., 2014*). The TFome was generated from a set of full cDNAs obtained from the Arizona Genomics Institute. The synthesis information, generated sequences, and request links for the maize TFome information are publicly available through GRASSIUS (https://grassius.org/tfomecollection.php). In conclusion, adaptation of available crop databases such as Gramene (*Tello-Ruiz et al., 2018*), and GRASSIUS for maize in machine learning approaches, as well as developing and adopting new databases, for example, the Wheat Information System (*Hu, Scheben & Edwards, 2018*) will help in the storage of data at the same time making it more accessible to scientists.

Traditional breeding techniques for selecting desirable traits depend on the observed phenotypic traits which can be misleading sometimes during selection, and this has led to low success in such strategies. Genomic selection is an accurate and efficient approach when it comes to the prediction of genotypic performance in crops. In maize, utilization of genomic techniques in the manipulation and analyses of TFs has been reported in a few

studies. For instance, *Shikha et al. (2017)* utilized genomic selection techniques on 240 subtropical maize lines during exposure to drought by selecting 29,619 SNPs. The study found that 77 out of 1,053 SNPs were linked with 10 drought associated TFs located inside a 150 kb area. For example, MYB (149 kb), WRKY (125 kb), NAC (149 kb), bZIP (92 kb), and AP-ERF (148 kb) among others. Similarly, transcriptome analysis of two maize inbred lines using RNA-Seq showed that a total of 2,558 and 555 genes responded to drought in both the sensitive and the tolerant lines, respectively (*Zhang et al., 2017a*). TFs were found to be enriched in the genotype-specific responsive genes, and the genotypic differentially expressed genes (DEGs). It was postulated that the differential expression of 22 TF genes and the genotype-specific response of 20 TFs in the tolerant line might play an important role in drought tolerance enhancement in maize. *Zhang et al. (2017b)* utilized RNA-Seq platform to analyze the expression of TFs in response to lead stress in a maize 178 lead-tolerant line. In this study, a total of 464 genes were expressed, among which 262 differentially expressed TFs which responded to Pb treatment were identified. More recently, (*Zhao et al., 2018*) mapped several abiotic stress responsive TFs to QTLs. The results showed that *MYB78* and *hsp70* were mapped to mQTL1-5. On chromosome 6, *pep7* and *mlip15* were both mapped on mQTL6-1. *Kusano et al. (1995)* demonstrated that *mlip15* was a low-temperature activated gene that encodes a bZIP protein made up of 135 amino acid. Elsewhere, global transcriptome profiling using RNA-seq on B73 maize seedlings exposed to heat, drought, cold stress, and salinity revealed 5,330 DEGs (*Li et al., 2017*). Functional annotations of these DEGs suggested that the pathways involving TFs, hormone metabolism and signaling among others controlled the stress responses. Among the 5,330 DEGs obtained, 167 genes were common to the four abiotic stresses; these included two down-regulated TFs (one MYB related and one b-ZIP) and 10 up-regulated TFs (one ARF, five ERFs, one MYB, one HD-ZIP, and two NAC). This study significantly contributes to a deeper understanding of molecular mechanisms involved in maize leaf responses to different abiotic stresses and could eventually contribute to the development of maize cultivars that are tolerant to various abiotic stresses.

Approaches involving genome-editing techniques create possibilities allowing for gene knockouts, point mutations, epigenetic changes, and the activation or repression of genes (*Kamburova et al., 2017*). *Svitashev et al. (2016)* reported the use of biolistic delivery of Cas9-gRNA ribonucleoproteins in maize cells, and this approach resulted in plants with both edited and mutated alleles. Recently (*Cox et al., 2017*), reported the use of single-effector programmable RNA guided RNases Cas13. This marks a major leap in plant transformation opening new opportunities in accelerated breeding in other main crops such as wheat, soybeans, sorghum, and rice. By using the RNA editing tools, the DNA structure is left intact but the function of the TF genes is manipulated. Utilization of genome editing techniques is still in its infancy and it is yet to be fully explored for abiotic stress tolerance in maize.

### Future outlook

Recent advances in genomics, molecular biology, metabolomics and proteomics have yielded fresh insights into the plant GRN, composed mainly of regulatory elements

(*trans*-elements and *cis*-elements), inducible genes (developmental and environmental cues), varying signal factors, and complementary biochemical pathways (*Tang, Harris & Newton, 2003*; *Wang et al., 2003*; *Zhu, 2002*). Sequencing of the whole maize genome has provided a basis for the functional characterization and identification of genetic networks and genes for maize improvement (*Schnable et al., 2009*). Moreover, recent availability of transcriptome profiling technologies, including genome sequencing and DNA microarrays, has opened new doors for understanding the patterns of transcription in the area of plant growth and development (*Sekhon et al., 2011*).

Understanding the genetic architecture of the molecular networks involved in maize, by utilizing current "OMICS" technologies is urgently needed to unravel the drought, heat and salt tolerance mechanisms in maize. Numerous genetic studies have shown that abiotic stress tolerance traits are usually polygenic making the selection of such traits extremely difficult (*Ciarmiello et al., 2011*). With the recent whole-genome sequencing of the B73 maize line, it's now feasible to identify most maize TFs and systematically estimate their contribution to abiotic stress tolerance. Maize has an increased level of genetic disequilibrium linkage (LD) and genetic diversity making it an ideal plant species since the LD and genetic diversity have been predicted to be within a number of kilobases (kbs) in maize landraces (*Tenaillon et al., 2001*). This characteristic makes genome-wide association studies (GWAS) at the gene level more accurate when compared to self-pollinated plant species as long as genome-wide and high-density DNA markers are present (*Yan, Warburton & Crouch, 2011*; *Li et al., 2012*). For example, using a Bayesian-based genome-wide association method in which RNA-seq-based systems of transcript buildup were utilized as explanatory variables (eRD-GWAS), genes linked to 13 traits were discovered from a group of 369 inbred maize lines (*Lin et al., 2017*). Additionally, TFs were found to be considerably enriched among the trait-associated genes discovered with eRD-GWAS. Similarly, genome-wide analyses carried out on the maize B73 inbred line to identify all the Hsf genes identified 25 non-redundant Hsf genes designated as *ZmHsfs* (*Lin et al., 2011*). In soybean, an all-inclusive phylogenetic study revealed 58 dehydration responsive genes from the *GmNAC* TF family (*Le et al., 2011*). RNA sequencing performed on 14-day old maize seedlings of inbred lines Mo17, B73, PH207, B37, and Oh43 under heat, cold, and control treatments, revealed a large number of genes that responded differentially between parental inbred lines (*Waters et al., 2017*). Moreover, 20 of the 57 annotated TF families in maize were enriched for elevated genes in heat and/or cold stress in at least three of the five inbred genotypes. Finally, TF families with TFs that were enriched for up-regulated genes in response to heat stress included MYB and HSF TF families, while 18 TF families with TFs enriched for up-regulated genes in response to cold stress included APETALA2/(AP2/EREB).

A new approach currently gaining rapid popularity is the field of phenomics. By utilizing high-throughput phenotyping, various physiological parameters such as biomass, internode length, leaf area, chlorophyll content, plant width and height, and growth rate can be accurately determined in real time, and noninvasively (*Rabara, Tripathi & Rushton, 2014*). Large amounts of quality phenomics data can be generated for many transgenic plants. Currently, field phenotyping systems are being developed with the
capacity to determine whether the engineering of TFs in plants can improve abiotic stress tolerance (*Rabara, Tripathi & Rushton, 2014*). For instance, *Awlia et al. (2016)* demonstrated that the phenotyping of polygenic traits in one experimental study could provide new insights into the mechanisms of plant responses to different abiotic stresses. The establishment of new phenomics technologies will further strengthen the use of forward genetics in the identification of novel TF genes regulating plant responses to different abiotic stresses.

Since TFs tend to regulate multiple pathways as opposed to a majority of the structural genes, they offer a powerful and unique system for use in the control of complex regulatory networks in plants. Overexpression of genes regulating the transcription of several down-stream abiotic/drought stress regulatory genes is a much better approach in the engineering of drought tolerant/resistant plants as opposed to the development of specific functional genes (*Bartels & Hussain, 2008*). Development of transgenic plants with enhanced abiotic stress tolerance by regulating TFs has become an important aspect of abiotic stress tolerance. Members of the WRKY, MYB, AREB, and bZIP, TF families have recently been utilized in the regulation of abiotic stress responses in major crops (*Singh, Foley & Oñate-Sánchez, 2002*). Many of the members belonging to these TF families have been identified and characterized in *Arabidopsis*, whose genome has been analyzed using microarray analysis, thus leading to the discovery of potential genes (*Shinozaki, Yamaguchi-Shinozaki & Seki, 2003*; *Bray, 2004*; *Denby & Gehring, 2005*). Thus, TF families offer important targets for use in gene manipulation and regulation which could be vital in understanding responses involved in abiotic stress tolerance. An increasing trend has seen the engineering of TFs involved in stress-signaling networks using biotechnology tools to generate transgenic stress tolerant plants. (*Abe et al., 2003*; *Sakuma et al., 2006*).

## CONCLUSION

The population in our planet is projected to rise to nine billion by the year 2050 (*Hussain, 2006*), together with the rapid changes in climate there is an urgent need to speed up the productivity of major crops. Understanding molecular mechanisms and mining stress-responsive genes that control plant responses to different abiotic stresses is a major prerequisite in the development of stress-resistant and high yielding crop varieties (*Khan et al., 2018*). To safeguard the global food production, crops (like maize) that are well adapted to adverse environmental conditions should be established (*Vinocur & Altman, 2005*; *M'mboyi et al., 2010*).

Maize is highly affected by abiotic stresses especially drought throughout its growth cycle, with the most damage being seen during the developmental stage and prior to flowering (*Claassen & Shaw, 1970*). TF mediated research in plants has recently shown progressive improvement since most of the TF encoding genes are early stress-responsive genes which control the expression of various downstream target genes (*Hoang et al., 2017*). This has in turn led to a deeper understanding of the involvement and functions of TFs in plant responses to different abiotic stresses (*Bartels & Sunkar, 2005*).

This review emphasizes on the main TF families and their potential in abiotic stress tolerance in maize. The majority of the TF genes in the literature are reported to play major roles in multiple abiotic stress tolerances. Among the target genes for engineering, the utilization of TFs has been recommended as they have potential to revolutionize biotechnology upon which novel crops with improved tolerance to abiotic stresses could be successfully generated. There is absence of literature available on abiotic stress responsive TFs with agronomic traits that have been utilized in maize in field conditions. Although Monsanto has developed, a biotechnology-derived inbred maize line that expresses HB17 (ATHB17), a TF from *Arabidopsis* (*Park et al., 2013*; *Hymus et al., 2013*). Expression of ATHB17 in the inbred line is linked with increased ear biomass at the silking stage compared to the near-isogenic controls (*Rice et al., 2014*). Increased ear biomass at the early stage of plant development is associated with increased silk size and greater grain yield from hybrid maize (*Borrás & Westgate, 2006*; *Lee & Tollenaar, 2007*).

Transcription factors are excellent candidates for the development of transgenic crops because of their roles in plant growth and development. Incorporation of abiotic stress response pathways in the vital reproductive and vegetative development stages in crops is an efficient strategy to improve productivity in field conditions (*Nelson et al., 2007*). TFs can be used to simulate a variety of developmental and biochemical networks that take part in the regulation of abiotic stresses, thus increasing the performance of crops in response to multiple plant abiotic stresses. *Joshi et al. (2016)* noted that overexpression of several TF genes significantly enhanced abiotic stress tolerance but at the same time caused a number of negative effects including lower yields, late flowering, and dwarfing in transgenic plants. This should be considered in future studies to maximize the effectiveness of TF engineering in responses to different abiotic stresses.

Moving forward, identification and characterization of multiple stress regulatory genes should be given more focus not only in maize but also in other major crops to target the most effective genes that can be universally used to develop abiotic stress tolerant crop varieties. Machine learning algorithms can be integrated with transcriptome data and high-throughput phenotyping data to further increase the automation of the gene discovery processes such as genome annotation and GRNs predictions.

Genetic engineering of multiple stress regulatory TF genes is a strong candidate for the enhancement of stress tolerance in plants when compared to focusing on a single individual gene. Nevertheless, recent advances in maize breeding, genomics, and functional analysis of genes combined with high-throughput sequencing technologies have significantly increased the chances of achieving multiple stress tolerances. The identification of commercial transgenic plants with enhanced crop performance under stress conditions is a tedious, expensive, and lengthy process. However, the successful genetic engineering of maize for improved abiotic stress tolerance using TFs as reviewed herein confirms this approach is feasible.

Since maize is a major crop in many countries, there is need for more collaboration in both applied and theoretical genomics in order to improve the production of maize. The rapid advancements in TFs genome analysis currently being witnessed are mostly on temperate maize varieties. It is hoped that these technologies can be transferred to

subtropical and tropical maize varieties that serve as essential food security crops in developing countries.

## ACKNOWLEDGEMENTS

The authors would like to sincerely thank the Center for Agricultural Resource Research, Institute of Genetics and Developmental biology (CAS) for availing the facilities.

### Funding

This work was generously supported by the National Key Research and Development Program of China (2016YFD0100102-11) and (2016YFD0100605) and the Project of the Innovative Academy of Seed Design, Chinese Academy of Sciences. The funders had no role in study design, data collection and analysis, decision to publish, or preparation of the manuscript.

### Grant Disclosures

The following grant information was disclosed by the authors:
National Key Research and Development Program of China: 2016YFD0100102-11 and 2016YFD0100605.
Project of the Innovative Academy of Seed Design, Chinese Academy of Sciences.

### Competing Interests

The authors declare that they have no competing interests.

### Author Contributions

- Roy Njoroge Kimotho conceived and designed the experiments, performed the experiments, analyzed the data, contributed reagents/materials/analysis tools, prepared figures and/or tables, authored or reviewed drafts of the paper, approved the final draft.
- Elamin Hafiz Baillo approved the final draft, revised the final paper.
- Zhengbin Zhang authored or reviewed drafts of the paper, approved the final draft, provided the framework and also revised the final draft.

### Data Availability

This article is a review article and does not have raw data.

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

# PeerJ

**Agalou A, Purwantomo S, Overnäs E, Johannesson H, Zhu X, Estiati A, De Kam RJ, Engström P, Slamet-Loedin IH, Zhu Z, Wang M, Xiong L, Meijer AH, Ouwerkerk PB. 2008.** A genome-wide survey of HD-Zip genes in rice and analysis of drought-responsive family members. *Plant Molecular Biology* **66(1–2)**:87–103 DOI 10.1007/s11103-007-9255-7.

**Ahuja I, De Vos RCH, Bones AM, Hall RD. 2010.** Plant molecular stress responses face climate change. *Trends in Plant Science* **15(12)**:664–674 DOI 10.1016/j.tplants.2010.08.002.

**Allan AC, Fluhr R. 2007.** Ozone and reactive oxygen species. Epub ahead of print 20 March 2017. *Encyclopedia of Life Sciences* DOI 10.1038/npg.els.0001299.

**Almoguera C, Rojas A, Díaz-Martín J, Prieto-Dapena P, Carranco R. 2002.** A seed-specific heat-shock transcription factor involved in developmental regulation during embryogenesis in sunflower. *Journal of Biological Chemistry* **277(46)**:43866–43872 DOI 10.1074/jbc.M207330200.

**Amajová O, Plíhal O, Al-Yousif M, Hirt H, Šamaj J. 2013.** Improvement of stress tolerance in plants by genetic manipulation of mitogen-activated protein kinases. *Biotechnology Advances* **31(1)**:118–128 DOI 10.1016/j.biotechadv.2011.12.002.

**Ariel FD, Manavella PA, Dezar CA, Chan RL. 2007.** The true story of the HD-Zip family. *Trends in Plant Science* **12(9)**:419–426 DOI 10.1016/j.tplants.2007.08.003.

**Ashburner M, Bonner JJ. 1979.** The induction of gene activity in drosophila by heat shock. *Cell* **17(2)**:241–254 DOI 10.1016/0092-8674(79)90150-8.

**Awlia M, Nigro A, Fajkus J, Schmoeckel SM, Negrão S, Santelia D, Trtílek M, Tester M, Julkowska MM, Panzarová K. 2016.** High-throughput non-destructive phenotyping of traits that contribute to salinity tolerance in *Arabidopsis thaliana*. *Frontiers in Plant Science* **7**:1414 DOI 10.3389/fpls.2016.01414.

**Banti V, Mafessoni F, Loreti E, Alpi A, Perata P. 2010.** The Heat-Inducible Transcription Factor *HsfA2* Enhances Anoxia Tolerance in *Arabidopsis*. *Plant Physiology* **152(3)**:1471–1483 DOI 10.1104/pp.109.149815.

**Bartels D, Hussain SS. 2008.** Current status and implications of engineering drought tolerance in plants using transgenic approaches. *CAB Reviews: Perspectives in Agriculture, Veterinary Science, Nutrition and Natural Resources* **3**:17.

**Bartels D, Sunkar R. 2005.** Drought and salt tolerance in plants. *Critical Reviews in Plant Sciences* **24(1)**:23–58 DOI 10.1080/07352680590910410.

**Bartoli CG, Casalongué CA, Simontacchi M, Marquez-Garcia B, Foyer CH. 2013.** Interactions between hormone and redox signalling pathways in the control of growth and cross tolerance to stress. *Environmental and Experimental Botany* **94**:73–88 DOI 10.1016/j.envexpbot.2012.05.003.

**Benatti P, Basile V, Merico D, Fantoni LI, Tagliafico E, Imbriano C. 2008.** A balance between NF-Y and *p53* governs the pro- and anti-apoptotic transcriptional response. *Nucleic Acids Research* **36(5)**:1415–1428 DOI 10.1093/nar/gkm1046.

**Bennetzen JL, Hake S. 2009.** *Handbook of maize: Genetics and genomics*. New York: Springer.

**Boch J, Scholze H, Schornack S, Landgraf A, Hahn S, Kay S, Lahaye T, Nickstadt A, Bonas U. 2009.** Breaking the code of DNA binding specificity of TAL-Type III Effectors. *Science* **326(5959)**:1509–1512 DOI 10.1126/science.1178811.

**Borrás L, Westgate ME. 2006.** Predicting maize kernel sink capacity early in development. *Field Crops Research* **95**:223–233.

**Bray EA. 2004.** Genes commonly regulated by water-deficit stress in *Arabidopsis thaliana*. *Journal of Experimental Botany* **55(407)**:2331–2341 DOI 10.1093/jxb/erh270.

**Burdo B, Gray J, Goetting-minesky MP, Wittler B, Hunt M, Li T, Velliquette D, Thomas J, Gentzel I, Dos Santos Brito M, Mejía-Guerra MK, Connolly LN, Qaisi D, Li W, Casas MI,**

**PeerJ** ______________________________________________________________

**Doseff AI, Grotewold E. 2014.** The Maize TFome development of a transcription factor open reading frame collection for functional genomics. *Plant Journal* **80(2)**:356–366 DOI 10.1111/tpj.12623.

**Butt HI, Yang Z, Chen E, Zhao G, Gong Q, Yang Z, Li F. 2017.** Functional characterization of cotton *GaMYB62L*, a novel *R2R3* TF in transgenic *Arabidopsis*. *PLOS ONE* **12(1)**:e0170578 DOI 10.1371/journal.pone.0170578.

**Cai R, Dai W, Zhang C, Wang Y, Wu M, Zhao Y, Ma Q, Xiang Y, Cheng B. 2017.** The maize WRKY transcription factor *ZmWRKY17* negatively regulates salt stress tolerance in transgenic *Arabidopsis* plants. *Planta* **246(6)**:1215–1231 DOI 10.1007/s00425-017-2766-9.

**Cai R, Zhao Y, Wang Y, Lin Y, Peng X. 2014.** Overexpression of a maize *WRKY58* gene enhances drought and salt tolerance in transgenic rice. *Plant Cell, Tissue and Organ Culture* **119(3)**:565–577 DOI 10.1007/s11240-014-0556-7.

**Cao L, Yu Y, Ding X, Zhu D, Yang F, Liu B, Sun X, Duan X, Yin K, Zhu Y. 2017.** The *Glycine soja* NAC transcription factor *GsNAC019* mediates the regulation of plant alkaline tolerance and ABA sensitivity. *Plant Molecular Biology* **95(3)**:253–268 DOI 10.1007/s11103-017-0643-3.

**Casaretto JA, El-Kereamy A, Zeng B, Stiegelmeyer SM, Chen X, Bi YM, Rothstein SJ. 2016.** Expression of *OsMYB55* in maize activates stress-responsive genes and enhances heat and drought tolerance. *BMC Genomics* **17(1)**:312 DOI 10.1186/s12864-016-2659-5.

**Chang W, Yin D. 2009.** Overexpression of maize *ZmDBP3* enhances tolerance to drought and cold stress in transgenic *Arabidopsis* plants. *Biologia* **64(6)**:1108–1114 DOI 10.2478/s11756-009-0198-0.

**Chávez Montes RA, Coello G, González-Aguilera KL, Marsch-Martínez N, De Folter S, Alvarez-Buylla ER. 2014.** ARACNe-based inference, using curated microarray data, of *Arabidopsis thaliana* root transcriptional regulatory networks. *BMC Plant Biology* **14(1)**:97 DOI 10.1186/1471-2229-14-97.

**Chaves MM, Oliveira MM. 2004.** Mechanisms underlying plant resilience to water deficits: prospects for water-saving agriculture. *Journal of Experimental Botany* **55(407)**:2365–2384 DOI 10.1093/jxb/erh269.

**Chen YH, Cao YY, Wang LJ, Li LM, Yang J, Zou MX. 2017.** Identification of MYB transcription factor genes and their expression during abiotic stresses in maize. *Biologia Plantarum* **62(2)**:222–230 DOI 10.1007/s10535-017-0756-1.

**Chen S, Huang X, Yan X, Liang Y, Wang Y, Li X, Peng X, Ma X, Zhang L, Cai Y, Ma T, Cheng L, Qi D, Zheng H, Yang X, Li X, Liu G. 2013.** Transcriptome analysis in sheepgrass (*Leymus chinensis*): a dominant perennial grass of the Eurasian Steppe. *PLOS ONE* **8(7)**:e67974 DOI 10.1371/journal.pone.0067974.

**Chen H, Lai Z, Shi J, Xiao Y, Chen Z, Xu X. 2010.** Roles of *ArabidopsisWRKY18, WRKY40* and *WRKY60* transcription factors in plant responses to abscisic acid and abiotic stress. *BMC Plant Biology* **10(1)**:281 DOI 10.1186/1471-2229-10-281.

**Chew W, Hrmova M, Lopato S. 2013.** Role of homeodomain leucine zipper (HD-Zip) iv transcription factors in plant development and plant protection from deleterious environmental factors. *International Journal of Molecular Sciences* **14(4)**:8122–8147 DOI 10.3390/ijms14048122.

**Christianson JA, Dennis ES, Llewellyn DJ, Wilson IW. 2010.** ATAF NAC transcription factors: regulators of plant stress signaling. *Plant Signaling & Behavior* **5(4)**:428–432 DOI 10.4161/psb.5.4.10847.

**Ciarmiello LF, Woodrow P, Fuggi A, Pontecorvo G, Carillo P. 2011.** Plant genes for abiotic stress. In: Shanker A, Venkateshwarlu B, eds. *Abiotic Stress in Plants—Mechanisms and*

*Adaptations*. London: InTech, 283–308. *Available at https://www.intechopen.com/books/abiotic-stress-in-plants-mechanisms-and-adaptations/plant-genes-for-abiotic-stress*.

**Claassen MM, Shaw RH. 1970.** Water deficit effects on corn. II Grain components. *Agronomy Journal* **62**:652–655.

**Cong L, Ran FA, Cox D, Lin S, Barretto R, Habib N, Hsu PD, Wu X, Jiang W, Marraffini LA, Zhang F. 2013.** Multiplex genome engineering using CRISPR/Cas systems. *Science* **339(6121)**:819–823 DOI 10.1126/science.1231143.

**Cox DBT, Gootenberg JS, Abudayyeh OO, Franklin B, Kellner MJ, Joung J, Zhang F. 2017.** RNA editing with CRISPR-Cas13. *Science* **358(6366)**:1019–1027 DOI 10.1126/science.aaq0180.

**Cui S, Youn E, Lee J, Maas SJ. 2014.** An improved systematic approach to predicting transcription factor target genes using support vector machine. *PLOS ONE* **9(4)**:e94519 DOI 10.1371/journal.pone.0094519.

**Cutler SR, Rodriguez PL, Finkelstein RR, Abrams SR. 2010.** Abscisic acid: emergence of a core signaling network. *Annual Reviews in Plant Biology* **61(1)**:651–679 DOI 10.1146/annurev-arplant-042809-112122.

**Dash S, Van Hemert J, Hong L, Wise RP, Dickerson JA. 2012.** PLEXdb: gene expression resources for plants and plant pathogens. *Nucleic Acids Research* **40(D1)**:D1194–D1201 DOI 10.1093/nar/gkr938.

**Denby K, Gehring C. 2005.** Engineering drought and salinity tolerance in plants: lessons from genome-wide expression profiling in *Arabidopsis*. *Trends in Biotechnology* **23(11)**:547–552 DOI 10.1016/j.tibtech.2005.09.001.

**Dixit S, Biswal AK, Min A, Henry A, Oane RH, Raorane ML, Longkumer T, Pabuayon IM, Mutte SK, Vardarajan AR, Miro B, Govindan G, Albano-Enriquez B, Pueffeld M, Sreenivasulu N, Slamet-Loedin I, Sundarvelpandian K, Tsai Y-C, Raghuvanshi S, Hsing Y-IC, Kumar A, Kohli A. 2015.** Action of multiple intra-QTL genes concerted around a co-localized transcription factor underpins a large effect QTL. *Scientific Reports* **5(1)**:15183 DOI 10.1038/srep15183.

**Dombrowski JE. 2003.** Salt stress activation of wound-related genes in tomato plants. *Plant Physiology* **132(4)**:2098–2107 DOI 10.1104/pp.102.019927.

**Dong W, Song Y, Zhao Z, Qiu NW, Liu X, Guo W. 2017.** The *Medicago truncatula* R2R3-MYB transcription factor gene *MtMYBS1* enhances salinity tolerance when constitutively expressed in *Arabidopsis thaliana*. *Biochemical and Biophysical Research Communications* **490(2)**:225–230 DOI 10.1016/j.bbrc.2017.06.025.

**Dong Y, Wang C, Han X, Tang S, Liu S, Xia X, Yin W. 2014.** A novel bHLH transcription factor *PebHLH35* from *Populus euphratica* confers drought tolerance through regulating stomatal development, photosynthesis and growth in *Arabidopsis*. *Biochemical and Biophysical Research Communications* **450(1)**:453–458 DOI 10.1016/j.bbrc.2014.05.139.

**Dou T, Hu C, Sun X, Shao X, Wu J, Ding L, Jie g, Wei H, Manosh B, Qiao Y, Yi G. 2015.** MpMYBS3 as a crucial transcription factor of cold signaling confers the cold tolerance of banana. *Plant Cell, Tissue and Organ Culture* **125(1)**:93–106 DOI 10.1007/s11240-015-0932-y.

**Du H, Wang YB, Xie Y, Liang Z, Jiang SJ, Zhang SS, Huang YB, Tang YX. 2013.** Genome-wide identification and evolutionary and expression analyses of MYB-related genes in land plants. *DNA Research* **20(5)**:437–448 DOI 10.1093/dnares/dst021.

**Echevarría-Zomeño S, Fernández-Calvino L, Castro-Sanz AB, López JA, Vázquez J, Castellano MM. 2016.** Dissecting the proteome dynamics of the early heat stress response leading to plant survival or death in *Arabidopsis*. *Plant, Cell & Environment* **39(6)**:1264–1278 DOI 10.1111/pce.12664.

**El-kereamy A, Bi Y-M, Ranathunge K, Beatty PH, Good AG, Rothstein SJ. 2012.** The rice R2R3-MYB transcription factor *OsMYB55* is involved in the tolerance to high temperature and modulates amino acid metabolism. *PLOS ONE* **7(12)**:e52030 DOI 10.1371/journal.pone.0052030.

**Faith JJ, Hayete B, Thaden JT, Mogno I, Wierzbowski J, Cottarel G, Kasif S, Collins JJ, Gardner TS. 2007.** Large-scale mapping and validation of *Escherichia coli* transcriptional regulation from a compendium of expression profiles. *PLOS Biology* **5(1)**:e8 DOI 10.1371/journal.pbio.0050008.

**Fornalé S, Shi X, Chai C, Encina A, Irar S, Capellades M, Fuguet E, Torres J-L, Rovira P, Puigdomenech P, Rigau J, Grotewold E, Gray J, Caparrós-Ruiz D. 2010.** *ZmMYB31* directly represses maize lignin genes and redirects the phenylpropanoid metabolic flux. *Plant Journal* **64(4)**:633–644 DOI 10.1111/j.1365-313X.2010.04363.x.

**Franco-Zorrilla JM, López-Vidriero I, Carrasco JL, Godoy M, Vera P, Solano R. 2014.** DNA-binding specificities of plant transcription factors and their potential to define target genes. *Proceedings of the National Academy of Sciences of the United States of America* **111(6)**:2367–2372 DOI 10.1073/pnas.1316278111.

**Gahlaut V, Jaiswal V, Kumar A, Gupta PK. 2016.** Transcription factors involved in drought tolerance and their possible role in developing drought tolerant cultivars with emphasis on wheat (*Triticum aestivum L.*). *Theoretical and Applied Genetics* **129(11)**:2019–2042 DOI 10.1007/s00122-016-2794-z.

**Gaj T, Gersbach CA, Barbas CF 3rd. 2013.** ZFN, TALEN, and CRISPR/Cas-based methods for genome engineering. *Trends in Biotechnology* **31(7)**:397–405 DOI 10.1016/j.tibtech.2013.04.004.

**Giraudat J, Parcy F, Bertauche N, Gosti F, Leung J, Morris P, Bouvier-Durand M, Vartanian N. 1994.** Current advances in abscisic acid action and signalling. *Plant Molecular Biology* **26(5)**:1557–1577.

**Gong F, Yang L, Tai F, Hu X, Wang W. 2014.** "Omics" of maize stress response for sustainable food production: opportunities and challenges. *OMICS: A Journal of Integrative Biology* **18(12)**:714–732 DOI 10.1089/omi.2014.0125.

**Gupta K, Jha B, Agarwal PK. 2014.** A dehydration-responsive element binding (DREB) transcription factor from the succulent halophyte *Salicornia brachiata* enhances abiotic stress tolerance in transgenic tobacco. *Marine Biotechnology* **16(6)**:657–673 DOI 10.1007/s10126-014-9582-z.

**Hoang XLT, Nhi DNH, Thu NBA, Thao NP, Tran LP. 2017.** Transcription factors and their roles in signal transduction in plants under abiotic stresses. *Current Genomics* **18(6)**:483–497 DOI 10.2174/1389202918666170227150057.

**Hu H, Dai M, Yao J, Xiao B, Li X, Zhang Q, Xiong L. 2006.** Overexpressing a NAM, ATAF, and CUC (NAC) transcription factor enhances drought resistance and salt tolerance in rice. *Proceedings of the National Academy of Sciences of the United States of America* **103(35)**:12987–12992 DOI 10.1073/pnas.0604882103.

**Hu H, Scheben A, Edwards D. 2018.** Advances in integrating genomics and bioinformatics in the plant breeding pipeline. *Agriculture* **8(6)**:75 DOI 10.3390/agriculture8060075.

**Hu H, You J, Fang Y, Zhu X, Qi Z, Xiong L. 2008.** Characterization of transcription factor gene *SNAC2* conferring cold and salt tolerance in rice. *Plant Molecular Biology* **67(1–2)**:169–181 DOI 10.1007/s11103-008-9309-5.

**Huang J, Zheng J, Yuan H, McGinnis K. 2018.** Distinct tissue-specific transcriptional regulation revealed by gene regulatory networks in maize. *BMC Plant Biology* **18(1)**:111 DOI 10.1186/s12870-018-1329-y.
**Hübel A, Schöffl F. 1994.** *Arabidopsis* heat shock factor: isolation and characterization of the gene and the recombinant protein. *Plant Molecular Biology* **26(1)**:353–362 DOI 10.1007/BF00039545.

**Hussain SS. 2006.** Molecular breeding for abiotic stress tolerance: drought perspective. *Proceedings of the Pakistan Academy* **43**:189–210.

**Hussain SS, Amjad M, Amjad M. 2011.** Transcription factors as tools to engineer enhanced drought stress tolerance in plants. *Biotechnology Progress* **27(2)**:297–306 DOI 10.1002/btpr.514.

**Hymus GJ, Cai S, Kohl EA, Holtan HE, Marion CM, Tiwari S, Maszle DR, Lundgren MR, Hong MC, Channa N, Loida P, Thompson R, Taylor JP, Rice E, Repetti PP, Ratcliffe OJ, Reuber TL, Creelman RA. 2013.** Application of HB17, an Arabidopsis class II homeodomain-leucine zipper transcription factor, to regulate chloroplast number and photosynthetic capacity. *Journal of Experimental Botany* **64(14)**:4479–4490 DOI 10.1093/jxb/ert261.

**Jaglo-Ottosen KR, Gilmour SJ, Zarka DG, Schabenberger O, Thomashow MF. 1998.** *Arabidopsis CBF1* overexpression induces *COR* genes and enhances freezing tolerance. *Science* **280(5360)**:104–106 DOI 10.1126/science.280.5360.104.

**Jakoby M, Weisshaar B, Dröge-Laser W, Vicente-Carbajosa J, Tiedemann J, Kroj T, Parcy F. 2002.** bZIP transcription factors in *Arabidopsis*. *Trends in Plant Science* **7(3)**:106–111 DOI 10.1016/s1360-1385(01)02223-3.

**Jia Z, Lian Y, Zhu Y, He J, Cao Z, Wang G. 2009.** Cloning and characterization of a putative transcription factor induced by abiotic stress in *Zea mays*. *African Journal of Biotechnology* **8(24)**:6764–6771.

**Jiang Y, Yang B, Harris NS, Deyholos MK. 2007.** Comparative proteomic analysis of NaCl stress-responsive proteins in *Arabidopsis* roots. *Journal of Experimental Botany* **58(13)**:3591–3607 DOI 10.1093/jxb/erm207.

**Jin C, Li K-Q, Xu X-Y, Zhang H-P, Chen H-X, Chen Y-H, Zhang S-L. 2017.** A novel NAC transcription factor *PbeNAC1*, of *Pyrus betulifolia* confers cold and drought tolerance via interacting with PbeDREBs and activating the expression of stress-responsive genes. *Frontiers in Plant Science* **8**:1049 DOI 10.3389/fpls.2017.01049.

**Jin J, Zhang H, Kong L, Gao G, Luo J. 2014.** Plant TFDB 3.0: a portal for the functional and evolutionary study of plant transcription factors. *Nucleic Acids Research* **42(D1)**:D1182–D1187 DOI 10.1093/nar/gkt1016.

**Johannesson H, Wang Y, Hanson J, Engstrom P. 2003.** The *Arabidopsis thaliana* homeobox gene *ATHB5* is a potential regulator of abscisic acid responsiveness in developing seedlings. *Plant Molecular Biology* **51(5)**:719–729.

**Joshi A, De Smet R, Marchal K, Van De Peer Y, Michoel T. 2009.** Module networks revisited: computational assessment and prioritization of model predictions. *Bioinformatics* **25(4)**:490–496 DOI 10.1093/bioinformatics/btn658.

**Joshi R, Wani SH, Singh B, Bohra A, Dar ZA, Lone AA, Pareek A, Singla-Pareek SL. 2016.** Transcription factors and plants response to drought stress: current understanding and future directions. *Frontiers in Plant Science* **7**:1029 DOI 10.3389/fpls.2016.01029.

**Kamburova VS, Nikitina EV, Shermatov SE, Buriev ZT, Kumpatla SP, Emani C, Abdurakhmonov IY. 2017.** Genome editing in plants: an overview of tools and applications. *International Journal of Agronomy* **2017**:1–15 DOI 10.1155/2017/7315351.

**Kasuga M, Liu Q, Miura S, Yamaguchi-Shinozaki K, Shinozaki K. 1999.** Improving plant drought, salt, and freezing tolerance by gene transfer of a single stress-inducible transcription factor. *Nature Biotechnology* **17(3)**:287–291 DOI 10.1038/7036.

**Khan S, Li M, Wang S, Yin H. 2018.** Revisiting the role of plant transcription factors in the battle against abiotic stress. *International Journal of Molecular Sciences* **19(6)**:1634 DOI 10.3390/ijms19061634.

**Kim YG, Cha J, Chandrasegaran S. 1996.** Hybrid restriction enzymes: zinc finger fusions to Fok I cleavage domain. *Proceedings of the National Academy of Sciences United States of America* **93(3)**:1156–1160 DOI 10.1073/pnas.93.3.1156.

**Kim SH, Hong JK, Lee SC, Sohn KH, Jung HW, Hwang BK. 2004.** *CAZFP1*, Cys2/His2-type zinc-finger transcription factor gene functions as a pathogen-induced early-defense gene in *Capsicum annuum*. *Plant Molecular Biology* **55(6)**:883–904 DOI 10.1007/s11103-005-2151-0.

**Kim C-Y, Vo KTX, Nguyen CD, Jeong D-H, Lee SK, Kumar M, Kim S-R, Park S-H, Kim J-K, Jeon J-S. 2016.** Functional analysis of a cold-responsive rice WRKY gene, *OsWRKY71*. *Plant Biotechnology Reports* **10(1)**:13–23 DOI 10.1007/s11816-015-0383-2.

**Kizis D, Pages M. 2002.** Maize DRE-binding proteins *DBF1* and *DBF2* are involved in *rab17* regulation through the drought-responsive element in an ABA-dependent pathway. *Plant Journal* **30(6)**:679–689 DOI 10.1046/j.1365-313x.2002.01325.x.

**Koryachko A, Matthiadis A, Ducoste JJ, Tuck J, Long TA, Williams C. 2015.** Computational approaches to identify regulators of plant stress response using high-throughput gene expression data. *Current Plant Biology* **3–4**:20–29 DOI 10.1016/j.cpb.2015.04.001.

**Kusano T, Berberich T, Harada M, Suzuki N, Sugawara K. 1995.** A maize DNA-binding factor with a bZIP motif is induced by low temperature. *MGG Molecular & General Genetics* **248(5)**:507–517 DOI 10.1007/bf02423445.

**Lata C, Yadav A, Prasad M. 2007.** Role of plant transcription factors in abiotic stress tolerance. Physiological, Biochemical and Genetic Perspectives. *Available at* https://www.intechopen.com/books/abiotic-stress-response-in-plants-physiological-biochemical-and-genetic-perspectives/role-of-plant-transcription-factors-in-abiotic-stress-tolerance.

**Le DT, Nishiyama R, Watanabe Y, Mochida K, Yamaguchi-Shinozaki K, Shinozaki K, Tran L-SP. 2011.** Genome-wide survey and expression analysis of the plant-specific NAC transcription factor family in soybean during development and dehydration stress. *DNA Research* **18(4)**:263–276 DOI 10.1093/dnares/dsr015.

**Lee Y-H, Oh H-S, Cheon C-I, Hwang I-T, Kim Y-J, Chun J-Y. 2001.** Structure and expression of the *Arabidopsis thaliana* homeobox gene *Athb-12*. *Biochemical and Biophysical Research Communications* **284(1)**:133–141 DOI 10.1006/bbrc.2001.4904.

**Lee EA, Tollenaar M. 2007.** Physiological basis of successful breeding strategies for maize grain yield. *Crop Science* **47(Supplement_3)**:S202–S215 DOI 10.2135/cropsci2007.04.0010ipbs.

**Li P, Cao W, Fang H, Xu S, Yin S, Zhang Y, Lin D, Wang J, Chen Y, Xu C, Yang Z. 2017.** Transcriptomic profiling of the maize (*Zea mays* L.) leaf response to abiotic stresses at the seedling stage. *Frontiers in Plant Science* **8**:290 DOI 10.3389/fpls.2017.00290.

**Li H, Gao Y, Xu H, Dai Y, Deng D, Chen J. 2013.** *ZmWRKY33*, a WRKY maize transcription factor conferring enhanced salt stress tolerances in *Arabidopsis*. *Plant Growth Regulation* **70(3)**:207–216 DOI 10.1007/s10725-013-9792-9.

**Li Z, Srivastava R, Tang J, Zheng Z, Howell SH. 2018.** Cis-effects condition the induction of a major unfolded protein response factor, *ZmbZIP60*, in response to heat stress in maize. *Frontiers in Plant Science* **9**:833 DOI 10.3389/fpls.2018.00833.

**Li Q, Yang X, Xu S, Cai Y, Zhang D, Han Y, Li L, Zhang Z, Gao S, Li J, Yan J. 2012.** Genome-wide association studies identified three independent polymorphisms associated with α-Tocopherol content in maize kernels. *PLOS ONE* **7(5)**:e36807 DOI 10.1371/journal.pone.0036807.

**Li H-C, Zhang H-N, Li G-L, Liu Z-H, Zhang Y-M, Zhang H-M, Guo X-L. 2015.** Expression of maize heat shock transcription factor gene *ZmHsf06* enhances the thermotolerance and drought-stress tolerance of transgenic *Arabidopsis*. *Functional Plant Biology* **42(11)**:1080–1091 DOI 10.1071/FP15080.

**Lin Z, Hong Y, Yin M, Li C, Zhang K, Grierson D. 2008.** A tomato HD-Zip homeobox protein, LeHB-1, plays an important role in floral organogenesis and ripening. *Plant Journal* **55(2)**:301–310 DOI 10.1111/j.1365-313x.2008.03505.x.

**Lin Y, Jiang H, Chu Z, Tang X, Zhu S, Cheng B. 2011.** Genome-wide identification, classification and analysis of heat shock transcription factor family in maize. *BMC Genomics* **12(1)**:76 DOI 10.1186/1471-2164-12-76.

**Lin H-Y, Liu Q, Li X, Yang J, Liu S, Huang Y, Schnable MJ, Nettleton D, Schnable PS. 2017.** Substantial contribution of genetic variation in the expression of transcription factors to phenotypic variation revealed by eRD-GWAS. *Genome Biology* **18(1)**:192 DOI 10.1186/s13059-017-1328-6.

**Liu L, Hao Z, Weng J, Li M, Zhang D, Bai L, Wang L, Li X, Zhang S. 2012.** Identification of drought-responsive genes by cDNA-amplified fragment length polymorphism in maize. *Annals of Applied Biology* **161(3)**:203–213 DOI 10.1111/j.1744-7348.2012.00565.x.

**Liu B, Hong Y-B, Zhang Y-F, Li X-H, Huang L, Zhang H-J, Li D-Y, Song F-M. 2014a.** Tomato WRKY transcriptional factor *SlDRW1* is required for disease resistance against *Botrytis cinerea* and tolerance to oxidative stress. *Plant Science* **227**:145–156 DOI 10.1016/j.plantsci.2014.08.001.

**Liu Q, Kasuga M, Sakuma Y, Abe H, Miura S, Yamaguchi-Shinozaki K, Shinozaki K. 1998.** Two transcription factors, DREB1 and DREB2, with an EREBP/AP2 DNA binding domain separate two cellular signal transduction pathways in drought- and low temperature-responsive gene expression, respectively, in *Arabidopsis*. *Plant Cell* **10(8)**:1391–1406 DOI 10.1105/tpc.10.8.1391.

**Liu C, Ruan Y, Lin Z, Wei R, Peng Q, Guan C, Ishii H. 2008.** Antagonism between acibenzolar-S-methyl-induced systemic acquired resistance and jasmonic acid-induced systemic acquired susceptibility to *Colletotrichum orbiculare* infection in cucumber. *Physiological and Molecular Plant Pathology* **72(4–6)**:141–145 DOI 10.1016/j.pmpp.2008.08.001.

**Liu TK, Song XM, Duan WK, Huang ZN, Liu GF, Li Y, Hou X. 2014b.** Genome-wide analysis and expression patterns of NAC transcription factor family under different developmental stages and abiotic stresses in Chinese Cabbage. *Plant Molecular Biology Reporter* **32(5)**:1041–1056 DOI 10.1007/s11105-014-0712-6.

**Liu S, Wang X, Wang H, Xin H, Yang X, Yan X, Li J, Tran L-SP, Shinozaki K, Yamaguchi-Shinozaki K, Qin F. 2013.** Genome-wide analysis of ZmDREB genes and their association with natural variation in drought tolerance at seedling stage of *Zea mays* L. *PLOS Genetics* **9(9)**:e1003790 DOI 10.1371/journal.pgen.1003790.

**Lowder LG, Zhou J, Zhang Y, Malzahn A, Zhong Z, Hsieh T-F, Voytas DF, Zhang Y, Qi Y. 2018.** Robust transcriptional activation in plants using multiplexed CRISPR-Act2.0 and mTALE-act systems. *Molecular Plant* **11(2)**:245–256 DOI 10.1016/j.molp.2017.11.010.

**Lu M, Sun QP, Zhang DF, Wang T, Pan T. 2015.** Identification of 7 stress related NAC transcription factor members in maize (*Zea mays* L.) and characterization of the expression pattern of these genes. *Biochemical and Biophysical Research Communications* **462(2)**:144–150 DOI 10.1016/j.bbrc.2015.04.113.

**Lu M, Ying S, Zhang D-F, Shi Y-S, Song Y-C, Wang T-Y, Li Y. 2012.** A maize stress-responsive NAC transcription factor, *ZmSNAC1*, confers enhanced tolerance to dehydration in transgenic *Arabidopsis*. *Plant Cell Reports* **31(9)**:1701–1711 DOI 10.1007/s00299-012-1284-2.

Luan M, Xu M, Lu Y, Zhang L, Fan Y, Wang L. 2014. Expression of zma-miR169 miRNAs and their target ZmNF-YA genes in response to abiotic stress in maize leaves. *Gene* **555(2)**:178–185 DOI 10.1016/j.gene.2014.11.001.

Ma H, Liu C, Li Z, Ran Q, Xie G, Wang B, Fang S, Chu J, Zhang J. 2018. *ZmbZIP4* contributes to stress resistance in maize by regulating ABA synthesis and root development. *Plant Physiology* **178(2)**:753–770 DOI 10.1104/pp.18.00436.

Ma X, Zhu X, Li C, Song Y, Zhang W, Xia G, Wang M. 2015. Overexpression of wheat *NF-YA10* gene regulates the salinity stress response in *Arabidopsis thaliana*. *Plant Physiology and Biochemistry* **86**:34–43 DOI 10.1016/j.plaphy.2014.11.011.

Mahalingam R. ed. 2015. "Consideration of combined stress: a crucial paradigm for improving multiple stress tolerance in plants". *Combined Stresses in Plants*. Cham: Springer International Publishing, 1–25.

Manavella PA, Arce AL, Dezar CA, Bitton F, Renou J-P, Crespi M, Chan RL. 2006. Cross-talk between ethylene and drought signalling pathways is mediated by the sunflower Hahb-4 transcription factor. *Plant Journal* **48(1)**:125–137 DOI 10.1111/j.1365-313x.2006.02865.x.

Mantovani R. 1999. The molecular biology of the CCAAT-binding factor NF-Y. *Gene* **239(1)**:15–27 DOI 10.1016/S0378-1119(99)00368-6.

Mao H, Wang H, Liu S, Li Z, Yang X, Yan J, Li J, Tran L-SP, Qin F. 2015. A transposable element in a NAC gene is associated with drought tolerance in maize seedlings. *Nature Communications* **6(1)**:8326 DOI 10.1038/ncomms9326.

Mao H, Yu L, Han R, Li Z, Liu H. 2016. *ZmNAC55*, a maize stress-responsive NAC transcription factor, confers drought resistance in transgenic *Arabidopsis*. *Plant Physiology and Biochemistry* **105**:55–66 DOI 10.1016/j.plaphy.2016.04.018.

Margolin AA, Nemenman I, Basso K, Wiggins C, Stolovitzky G, Favera RD, Califano A. 2006. ARACNE: an algorithm for the reconstruction of gene regulatory networks in a mammalian cellular context. *BMC Bioinformatics* **7(Suppl 1)**:S7 DOI 10.1186/1471-2105-7-S1-S7.

Misra A, Sriram G. 2013. Network component analysis provides quantitative insights on an Arabidopsis transcription factor-gene regulatory network. *BMC Systems Biology* **7(1)**:126 DOI 10.1186/1752-0509-7-126.

Mittler R. 2006. Abiotic stress, the field environment and stress combination. *Trends in Plant Science* **11(1)**:15–19 DOI 10.1016/j.tplants.2005.11.002.

M'mboyi F, Mugo S, Mwimali M, Ambani L. 2010. *Maize production and improvement in Sub-Saharan Africa*. Nairobi: African Biotechnology Stakeholders Forum.

Moon S-J, Han S-Y, Kim D-Y, Yoon IS, Shin D, Byun M-O, Kwon H-B, Kim B-G. 2015. Ectopic expression of a hot pepper bZIP-like transcription factor in potato enhances drought tolerance without decreasing tuber yield. *Plant Molecular Biology* **89(4–5)**:421–431 DOI 10.1007/s11103-015-0378-y.

Nakashima K, Ito Y, Yamaguchi-Shinozaki K. 2009. Transcriptional regulatory networks in response to abiotic stresses in *Arabidopsis* and Grasses. *Plant Physiology* **149(1)**:88–95 DOI 10.1104/pp.108.129791.

Nardini M, Gnesutta N, Donati G, Gatta R, Forni C, Fossati A, Vonrhein C, Moras D, Romier M, Matovani R. 2013. Sequence-specific transcription factor NF-Y displays histone-like DNA binding and H2B-like ubiquitination. *Cell* **152(1–2)**:132–143 DOI 10.1016/j.cell.2012.11.047.

Nelson DE, Repetti PP, Adams TR, Creelman RA, Wu J, Warner DC, Anstrom DC, Bensen RJ, Castiglioni PP, Donnarummo MG, Hinchey BS, Kumimoto RW, Maszle DR, Canales RD, Krolikowski KA, Dotson SB, Gutterson N, Ratcliffe OJ, Heard JE. 2007. Plant nuclear factor Y (NF-Y) B subunits confer drought tolerance and lead to improved corn yields on

water-limited acres. *Proceedings of the National Academy of Sciences of the United States of America* **104(42)**:16450–16455 DOI 10.1073/pnas.0707193104.

**Nepolean T, Kaul J, Mukri G, Mittal S. 2018.** Genomics-enabled next-generation breeding approaches for developing system-specific drought tolerant hybrids in maize. *Frontiers in Plant Science* **9**:361 DOI 10.3389/fpls.2018.00361.

**Ni Y, Aghamirzaie D, Elmarakeby H, Collakova E, Li S, Grene R, Heath LS. 2016.** A machine learning approach to predict gene regulatory networks in seed development in *Arabidopsis*. *Frontiers in Plant Science* **7**:1936 DOI 10.3389/fpls.2016.01936.

**Nuruzzaman M, Manimekalai R, Sharoni AM, Satoh K, Kondoh H, Ooka H, kikuchi S. 2010.** Genome-wide analysis of NAC transcription factor family in rice. *Gene* **465(1–2)**:30–44 DOI 10.1016/j.gene.2010.06.008.

**Ogawa O, Yamaguchi K, Nishiuchi T. 2007.** High-level overexpression of the *Arabidopsis HsfA2* gene confers not only increased themotolerance but also salt/osmotic stress tolerance and enhanced callus growth. *Journal of Experimental Botany* **58(12)**:3373–3383 DOI 10.1093/jxb/erm184.

**Ooka H, Satoh K, Doi K, Nagata T, Otomo Y, Murakami K, Matsubara K, Osato N, Kawai J, Carninci P, Hayashizaki Y, Suzuki K, Kojima K, Takahara Y, Yamamoto K, Kikuchi S. 2003.** Comprehensive analysis of NAC family genes in *Oryza sativa* and *Arabidopsis thaliana*. *DNA Research* **10(6)**:239–247 DOI 10.1093/dnares/10.6.239.

**Pandey P, Ramegowda V, Senthil-Kumar M. 2015.** Shared and unique responses of plants to multiple individual stresses and stress combinations: physiological and molecular mechanisms. *Frontiers in Plant Science* **6**:723 DOI 10.3389/fpls.2015.00723.

**Park MY, Kim SA, Lee SJ, Kim SY. 2013.** ATHB17 is a positive regulator of abscisic acid response during early seedling growth. *Molecules and Cells* **35(2)**:125–133 DOI 10.1007/s10059-013-2245-5.

**Paz-Ares J, Ghosal D, Wienand U, Peterson PA, Saedler H. 1987.** The regulatory cl locus of *Zea mays* encodes a protein with homology to myb proto-oncogene products and with structural similarities to transcriptional activators. *EMBO Journal* **6(12)**:3553–3558 DOI 10.1002/j.1460-2075.1987.tb02684.x.

**Penfold CA, Wild DL. 2011.** How to infer gene networks from expression profiles, revisited. *Interface Focus* **1**:857–870 DOI 10.1098/rsfs.2011.0053.

**Pérez-Rodríguez P, Riaño-Pachón DM, Corrêa LGG, Rensing SA, Kersten B, Mueller-Roeber B. 2010.** PlnTFDB: updated content and new features of the plant transcription factor database. *Nucleic Acids Research* **38**:D822–D8227 DOI 10.1093/nar/gkp805.

**Perlack RD, Wright LL, Turhollow AF, Graham RL, Stokes BJ, Erbach DC. 2005.** *Biomass as feedstock for a bioenergy and bioproducts industry: The technical feasibility of a billion-ton annual supply*. Oak Ridge: Oak Ridge National Laboratory.

**Phukan UJ, Jeena GS, Shukla RK. 2016.** WRKY transcription factors: molecular regulation and stress responses in plants. *Frontiers in Plant Science* **7**:760 DOI 10.3389/fpls.2016.00760.

**Phukan UJ, Mishra S, Timbre K, Luqman S, Shukla RK. 2014.** *Mentha arvensis* exhibit better adaptive characters in contrast to *Mentha piperita* when subjugated to sustained water logging stress. *Protoplasma* **251(3)**:603–614 DOI 10.1007/s00709-013-0561-4.

**Prasad PVV, Pisipati SR, Momcilovic I, Ristic Z. 2011.** Independent and combined effects of high temperature and drought stress during grain filling on plant yield and chloroplast EF-Tu expression in spring wheat. *Journal of Agronomy and Crop Science* **197(6)**:430–441 DOI 10.1111/j.1439-037X.2011.00477.x.

Qin F, Kakimoto M, Sakuma Y, Maruyama K, Osakabe Y, Tran L-SP, Yamaguchi-Shinozaki K. 2007. Regulation and functional analysis of ZmDREB2A in response to drought and heat stresses in *Zea mays* L. *Plant Journal* **50(1)**:54–69 DOI 10.1111/j.1365-313x.2007.03034.x.

Qin F, Sakuma Y, Li J, Liu Q, Li Y, Shinozaki K, Yamaguchi-Shinozaki K. 2004. Cloning and functional analysis of a novel *DREB1/CBF* transcription factor involved in cold-responsive gene expression in *Zea mays* L. *Plant Cell Physiology* **45(8)**:1042–1052 DOI 10.1093/pcp/pch118.

Qing M, Wei D. 2018. Functional analysis of mam-resistant candidate gene *Zmhdz12*. Beijing: China Science and Technology Paper Online [2017-04-26]. *Available at* http://www.doc88.com/p-9701334256860.html.

Rabara RC, Tripathi P, Rushton PJ. 2014. The potential of transcription factor-based genetic engineering in improving crop tolerance to drought. *Omics: A Journal of Integrative Biology* **18(10)**:601–614 DOI 10.1089/omi.2013.0177.

Ramakrishna C, Singh S, Raghavendrarao S, Padaria JC, Mohanty S, Sharma TR, Solanke AU. 2018. The membrane tethered transcription factor *EcbZIP17* from finger millet promotes plant growth and enhances tolerance to abiotic stresses. *Scientific Reports* **8(1)**:2148 DOI 10.1038/s41598-018-19766-4.

Ramegowda V, Senthil-Kumar M. 2015. The interactive effects of simultaneous biotic and abiotic stresses on plants: mechanistic understanding from drought and pathogen combination. *Journal of Plant Physiology* **176**:47–54 DOI 10.1016/j.jplph.2014.11.008.

Rasmussen S, Barah P, Suarez-Rodriguez MC, Bressendorff S, Friis P, Costantino P, Bones AM, Nielsen HB, Mundy J. 2013. Transcriptome responses to combinations of stresses on Arabidopsis. *Plant Physiology* **161(4)**:1783–1794 DOI 10.1104/pp.112.210773.

Renau-Morata B, Molina RV, Carrillo L, Cebolla-Cornejo J, Sánchez-Perales M, Pollmann S, Domínguez-Figueroa J, Corrales AR, Flexas J, Vicente-Carbajosa J, Medina J, Nebauer SG. 2017. Ectopic expression of CDF3 genes in tomato enhances biomass production and yield under salinity stress conditions. *Frontiers in Plant Science* **8**:660 DOI 10.3389/fpls.2017.00660.

Riechmann JL, Heard J, Martin G, Reuber L, Jiang CZ, Keddie J, Adam L, Pineda O, Ratcliffe OJ, Samaha RR, Crelman R, Pilgrim M, Broun P, Zhang JZ, Ghandehari D, Sherman BK, Yu GL. 2000. *Arabidopsis* transcription factors: genome-wide comparative analysis among eukaryotes. *Science* **290(5499)**:2105–2110 DOI 10.1126/science.290.5499.2105.

Riechmann JL, Meyerowitz EM. 1998. The AP2/EREBP family of plant transcription factors. *Biological Chemistry* **379(6)**:633–646.

Rice EA, Khandelwal A, Creelman RA, Griffith C, Ahrens JE, Taylor JP, Murphy LR, Manjunath S, Thompson RL, Lingard MJ, Back SL, Larue H, Brayton BR, Burek AJ, Tiwari S, Adam L, Morrell JA, Caldo RA, Huai Q, Kouadio J-lK, Kuehn R, Sant AM, Wingbermuehle WJ, Sala R, Foster M, Kinser JD, Mohanty R, Jiang D, Ziegler TE, Huang MG, Kuriakose SV, Skottke K, Repetti PP, Reuber TL, Ruff TG, Petracek ME, Loida PJ. 2014. Expression of a truncated ATHB17 protein in maize increases ear weight at silking. *PLOS ONE* **9(4)**:e94238 DOI 10.1371/journal.pone.0094238.

Rushton PJ, Bokowiec MT, Han S, Zhang H, Brannock JF, Chen X, Laudeman TW, Timko MP. 2008. Tobacco transcription factors: novel insights into transcriptional regulation in the Solanaceae. *Plant Physiology* **147(1)**:280–295 DOI 10.1104/pp.107.114041.

Rushton PJ, Somssich IE, Ringler P, Shen QXJ. 2010. WRKY transcription factors. *Trends in Plant Science* **15(5)**:247–258 DOI 10.1016/j.tplants.2010.02.006.

Saibo NJM, Lourenço T, Oliveira MM. 2009. Transcription factors and regulation of photosynthetic and related metabolism under environmental stresses. *Annals of Botany* **103(4)**:609–623 DOI 10.1093/aob/mcn227.
**Sakuma Y, Maryyama K, Qin F, Osakabe Y, Shinozaki K, Yamaguchi-Shinozaki K. 2006.**
Dual function of an *Arabidopsis* transcription factor *DREB2A* in water-stress-responsive and heat-stress-responsive gene expression. *Proceedings of the National Academy of Sciences of the United States of America* **103(49)**:18822–18827 DOI 10.1073/pnas.0605639103.

**Saleh A, Lumreras V, Pages M. 2005.** Functional role of DRE binding transcription factors in abiotic stress. In: Tuberosa R, Phillips RL, Gale M, eds. *Proceedings of the International Congress 'In the Wake of the Double Helix From the Green Revolution to the Gene Revolution'. 27–31 May 2003.* Bologna: Avenue Media, 193–205.

**Sato H, Mizoi J, Tanaka H, Maruyama K, Qin F, Osakabe Y, Morimoto K, Ohori T, Kusakabe K, Nagata M, Shinozaki K, Yamaguchi-Shinozaki K. 2014.** *Arabidopsis* DPB3-1, a DREB2A interactor, specifically enhances heat stress induced gene expression by forming a heat stress-specific transcriptional complex with NF-Y subunits. *Plant Cell* **26(12)**:4954–4973 DOI 10.1105/tpc.114.132928.

**Scharf K, Berberich T, Ebersberger I, Nover L. 2012.** Plant heat stress transcription factor (Hsf) family: structure, function and evolution. *Biochimica et Biophysica Acta (BBA)—Gene Regulatory Mechanisms* **1819(2)**:104–119 DOI 10.1016/j.bbagrm.2011.10.002.

**Scharf KD, Rose S, Zott W, Schöffl F, Nover L. 1990.** Three tomato genes code for heat stress transcription factors with a region of remarkable homology to the DNA-binding domain of the yeast HSF. *EMBO Journal* **9(13)**:4495–4501 DOI 10.1002/j.1460-2075.1990.tb07900.x.

**Schnable PS, Ware D, Fulton RS, Stein JC, Wei F, Pasternak S, Liang C, Zhang J, Fulton L, Graves TA, Minx P, Reily AD, Courtney L, Kruchowski SS, Tomlinson C, Strong C, Delehaunty K, Fronick C, Courtney B, Rock SM, Belter E, Du F, Kim K, Abbott RM, Cotton M, Levy A, Marchetto P, Ochoa K, Jackson SM, Gillam B, Chen W, Yan L, Higginbotham J, Cardenas M, Waligorski J, Applebaum E, Phelps L, Falcone J, Kanchi K, Thane T, Scimone A, Thane N, Henke J, Wang T, Ruppert J, Shah N, Rotter K, Hodges J, Ingenthron E, Cordes M, Kohlberg S, Sgro J, Delgado B, Mead K, Chinwalla A, Leonard S, Crouse K, Collura K, Kudrna D, Currie J, He R, Angelova A, Rajasekar S, Mueller T, Lomeli R, Scara G, Ko A, Delaney K, Wissotski M, Lopez G, Campos D, Braidotti M, Ashley E, Golser W, Kim H, Lee S, Lin J, Dujmic Z, Kim W, Talag J, Zuccolo A, Fan C, Sebastian A, Kramer M, Spiegel L, Nascimento L, Zutavern T, Miller B, Ambroise C, Muller S, Spooner W, Narechania A, Ren L, Wei S, Kumari S, Faga B, Levy MJ, McMahan L, Van Buren P, Vaughn MW, Ying K, Yeh CT, Emrich SJ, Jia Y, Kalyanaraman A, Hsia AP, Barbazuk WB, Baucom RS, Brutnell TP, Carpita NC, Chaparro C, Chia JM, Deragon JM, Estill JC, Fu Y, Jeddeloh JA, Han Y, Lee H, Li P, Lisch DR, Liu S, Liu Z, Nagel DH, McCann MC, SanMiguel P, Myers AM, Nettleton D, Nguyen J, Penning BW, Ponnala L, Schneider KL, Schwartz DC, Sharma A, Soderlund C, Springer NM, Sun Q, Wang H, Waterman M, Westerman R, Wolfgruber TK, Yang L, Yu Y, Zhang L, Zhou S, Zhu Q, Bennetzen JL, Dawe RK, Jiang J, Jiang N, Presting GG, Wessler SR, Aluru S, Martienssen RA, Clifton SW, McCombie WR, Wing RA, Wilson RK. 2009.** The B73 maize genome: complexity, diversity, and dynamics. *Science* **326(5956)**:1112–1115 DOI 10.1126/science.1178534.

**Schuetz TJ, Gallo GJ, Sheldon L, Tempst P, Kingston RE. 1991.** Isolation of a cDNA for HSF2: evidence for two heat shock factor genes in humans. *Proceedings of the National Academy of Sciences of the United States of America* **88(16)**:6911–6915 DOI 10.1073/pnas.88.16.6911.

**Sekhon RS, Lin H, Childs KL, Hansey CN, Buell CR, De Leon N, Kaeppler SM. 2011.** Genome-wide atlas of transcription during maize development. *Plant Journal* **66(4)**:553–563 DOI 10.1111/j.1365-313X.2011.04527.

**Shang Y, Yan L, Liu Z-Q, Cao Z, Mei C, Xin Q, Wu F-Q, Wang X-F, Du S-Y, Jiang T, Zhang X-F, Zhao R, Sun H-L, Liu R, Yu Y-T, Zhang D-P. 2010.** The Mg-chelatase H subunit of *Arabidopsis*

antagonizes a group of WRKY transcription repressors to relieve ABA-responsive genes of inhibition. *Plant Cell* **22(6)**:1909–1935 DOI 10.1105/tpc.110.073874.

**Shao H, Wang H, Tang X. 2015.** NAC transcription factors in plant multiple abiotic stress responses: progress and prospects. *Frontiers in Plant Science* **6**:902 DOI 10.3389/fpls.2015.00902.

**Sharoni AM, Nuruzzaman M, Satoh K, Shimizu T, Kondoh H, Sasaya T, Choi IR, Omura T, Kikuchi S. 2011.** Gene structures, classification and expression models of the AP2/EREBP transcription factor family in rice. *Plant and Cell Physiology* **52(2)**:344–360 DOI 10.1093/pcp/pcq196.

**Shen Q, Zhang P, Ho T-HD. 1996.** Modular nature of abscisic acid (ABA) response complexes: composite promoter units that are necessary and sufficient for ABA induction of gene expression in barley. *Plant Cell* **8(7)**:1107–1119 DOI 10.2307/3870355.

**Shikha M, Kanika A, Rao AR, Mallikarjuna MG, Gupta HS, Nepolean T. 2017.** Genomic selection for drought tolerance using genome-wide SNPs in maize. *Frontiers in Plant Science* **8**:550 DOI 10.3389/fpls.2017.00550.

**Shim D, Hwang J-U, Lee J, Lee S, Choi Y, An G, Martinoia E, Lee Y. 2009.** Orthologs of the class A4 heat shock transcription factor *HSFA4a* confer cadmium tolerance in wheat and rice. *Plant Journal* **21(12)**:4031–4043 DOI 10.1105/tpc.109.066902.

**Shinozaki K, Yamaguchi-Shinozaki K. 2006.** Gene networks involved in drought stress response and tolerance. *Journal of Experimental Botany* **58(2)**:221–227 DOI 10.1093/jxb/erl164.

**Shinozaki K, Yamaguchi-Shinozaki K, Seki M. 2003.** Regulatory network of gene expression in the drought and cold stress responses. *Current Opinion in Plant Biology* **6(5)**:410–417 DOI 10.1016/s1369-5266(03)00092-x.

**Shiriga K, Sharma R, Kumar K, Yadav SK, Hossain F, Thirunavukkarasu N. 2014.** Genome-wide identification and expression pattern of drought-responsive members of the NAC family in maize. *Meta Gene* **2**:407–417 DOI 10.1016/j.mgene.2014.05.001.

**Siefers N, Dang KK, Kumimoto RW, Bynum WE, Tayrose G, Holt BF. 2009.** Tissue-specific expression patterns of *Arabidopsis* NF-Y transcription factors suggest potential for extensive combinatorial complexity. *Plant Physiology* **149(2)**:625–641 DOI 10.1104/pp.108.130591.

**Singh K, Foley RC, Oñate-Sánchez L. 2002.** Transcription factors in plant defense and stress responses. *Current Opinion in Plant Biology* **5(5)**:430–436 DOI 10.1016/s1369-5266(02)00289-3.

**Soderman E, Hjellstrom M, Fahleson J, Engstrom P. 1999.** The HD-Zip gene *ATHB6* in *Arabidopsis* is expressed in developing leaves, roots and carpels and up-regulated by water deficit conditions. *Plant Molecular Biology* **40(6)**:1073–1083.

**Soderman E, Mattsson J, Engstrom P. 1996.** The *Arabidopsis* homeobox gene *ATHB-7* is induced by water deficit and by abscisic acid. *Plant Journal* **10(2)**:375–381 DOI 10.1046/j.1365-313x.1996.10020375.x.

**Song X, Li Y, Hou X. 2013.** Genome-wide analysis of the AP2/ERF transcription factor super family in Chinese cabbage (*Brassicarapa ssp. pekinensis*). *BMC Genomics* **14(1)**:573 DOI 10.1186/1471-2164-14-573.

**Song J, Weng Q, Ma H, Yuan J, Wang L, Liu Y. 2016.** Cloning and expression analysis of the Hsp70 gene *ZmERD2* in *Zea mays*. *Biotechnology & Biotechnological Equipment* **30(2)**:219–226 DOI 10.1080/13102818.2015.1131625.

**Su H, Cao Y, Ku L, Yao W, Cao Y, Ren Z, Dou D, Wang H, Ren Z, Liu H, Tian L, Zheng Y, Chen C, Chen Y. 2018.** Dual functions of *ZmNF-YA3* in photoperiod-dependent flowering and abiotic stress responses in maize. *Journal of Experimental Botany* **69(21)**:5177–5189 DOI 10.1093/jxb/ery299.

**Sun X-C, Gao Y-F, Li H-R, Yang S-Z, Liu Y-S. 2015.** Over-expression of *SlWRKY39* leads to enhanced resistance to multiple stress factors in tomato. *Journal of Plant Biology* **58(1)**:52–60 DOI 10.1007/s12374-014-0407-4.

**Sun H, Huang X, Xu X, Lan H, Huang J, Zhang H-S. 2012.** ENAC1, a NAC transcription factor, is an early and transient response regulator induced by abiotic stress in rice (*Oryza sativa* L.). *Molecular Biotechnology* **52(2)**:101–110 DOI 10.1007/s12033-011-9477-4.

**Suzuki N, Rivero RM, Shulaev V, Blumwald E, Mittler R. 2014.** Abiotic and biotic stress combinations. *New Phytologist* **203(1)**:32–43 DOI 10.1111/nph.12797.

**Svitashev S, Schwartz C, Lenderts B, Young JK, Cigan AM. 2016.** Genome editing in maize directed by CRISPR–Cas9 ribonucleoprotein complexes. *Nature Communications* **7(1)**:13274 DOI 10.1038/ncomms13274.

**Tang W, Harris L, Newton RJ. 2003.** Molecular mechanism of salinity stress and biotechnological strategies for engineering salt tolerance in plants. *Forestry Studies in China* **5(2)**:52–62.

**Tello-Ruiz MK, Naithani S, Stein JC, Gupta P, Campbell M, Olson A, Wei S, Preece J, Geniza MJ, Jiao Y, Lee YK, Wang B, Mulvaney J, Chougule K, Elser J, Al-Bader N, Kumari S, Thomason J, Kumar V, Bolser DM, Naamati G, Tapanari E, Fonseca N, Huerta L, Iqbal H, Keays M, Munoz-Pomer Fuentes A, Tang A, Fabregat A, D'Eustachio P, Weiser J, Stein LD, Petryszak R, Papatheodorou I, Kersey PJ, Lockhart P, Taylor C, Jaiswal P, Ware D. 2018.** Gramene 2018: unifying comparative genomics and pathway resources for plant research. *Nucleic Acids Research* **46(D1)**:D1181–D1189 DOI 10.1093/nar/gkx1111.

**Tenaillon MI, Sawkins MC, Long AD, Gaut RL, Doebley JF, Gaut BS. 2001.** Patterns of DNA sequence polymorphism along chromosome 1 of maize (*Zea mays ssp. mays L*). *Proceedings of the National Academy of Sciences of the United States of America* **98(16)**:9161–9166 DOI 10.1073/pnas.151244298.

**Todaka D, Shinozaki K, Yamaguchi-Shinozaki K. 2015.** Recent advances in the dissection of drought-stress regulatory networks and strategies for development of drought-tolerant transgenic rice plants. *Frontiers in Plant Science* **6**:84 DOI 10.3389/fpls.2015.00084.

**Tran L-SP, Nakashima K, Sakuma Y, Osakabe Y, Qin F, Simpson SD, Maruyama K, Fujita Y, Shinozaki K, Yamaguchi-Shinozaki K. 2007.** Co-expression of the stress-inducible zinc finger homeodomain *ZFHD1* and NAC transcription factors enhances expression of the *ERD1* gene in *Arabidopsis*. *Plant Journal* **49(1)**:46–63 DOI 10.1111/j.1365-313x.2006.02932.x.

**Tran L-SP, Nakashima K, Sakuma Y, Simpson SD, Fujita Y, Maruyama K, Fujita M, Seki M, Shinozaki K, Yamaguchi-Shinozaki K. 2004.** Isolation and Functional Analysis of Arabidopsis Stress-Inducible NAC Transcription Factors That Bind to a Drought-Responsive *cis*-Element in the early responsive to dehydration stress 1 Promoter. *The Plant Cell* **16(9)**:2481–2498 DOI 10.1105/tpc.104.022699.

**Tripathi P, Rabara RC, Rushton PJ. 2014.** A systems biology perspective on the role of WRKY transcription factors in drought responses in plants. *Planta* **239(2)**:255–266 DOI 10.1007/s00425-013-1985-y.

**Ulker B, Somssich IE. 2004.** WRKY transcription factors: from DNA binding towards biological function. *Current Opinion in Plant Biology* **7(5)**:491–498 DOI 10.1016/j.pbi.2004.07.012.

**Ullah A, Sun H, Hakim, Yang X, Zhang X. 2017.** A novel cotton WRKY gene, *GhWRKY6*-like, improves salt tolerance by activating the ABA signalling pathway and scavenging of reactive oxygen species. *Physiologia Plantarum* **162(4)**:439–454 DOI 10.1111/ppl.12651.

**Umezawa T, Fujita M, Fujita Y, Yamaguchi-Shinozaki K, Shinozaki K. 2006.** Engineering drought tolerance in plants: discovering and tailoring genes to unlock the future. *Current Opinion in Biotechnology* **17(2)**:113–122 DOI 10.1016/j.copbio.2006.02.002.

**Vermeirssen V, De Clercq I, Van Parys T, Van Breusegem F, Van De Peer Y. 2014.** *Arabidopsis* ensemble reverse-engineered gene regulatory network discloses interconnected transcription factors in oxidative stress. *Plant Cell* **26(12)**:4656–4679 DOI 10.1105/tpc.114.131417.

**Vinocur B, Altman A. 2005.** Recent advances in engineering plant tolerance to abiotic stress: Achievements and limitations. *Current Opinion in Biotechnology* **16(2)**:123–132 DOI 10.1016/j.copbio.2005.02.001.

**Wang N. 2014.** Analysis of abiotic stress related functions of genes *ZmWRKY50* and *ZmWRKY44* in Maize (*Zea Mays L.*). Master's thesis. Sichuan Agricultural University, Sichuan Sheng, China.

**Wang WX, Barak T, Vinocur B, Shoseyov O, Altman A. 2003.** Abiotic resistance and chaperones: possible physiological role of SP1, a stable and stabilizing protein from Populus. In: Vasil IK, ed. *Plant Biotechnology 2000*. Dordrecht: Kluwer, 439–443.

**Wang C-T, Dong Y-M. 2009.** Overexpression of maize ZmDBP3 enhances tolerance to drought and cold stress in transgenic Arabidopsis plants. *Biologia* **64(6)** DOI 10.2478/s11756-009-0198-0.

**Wang B, Li Z, Ran Q, Li P, Peng Z, Zhang J. 2018b.** *ZmNF-YB16* overexpression improves drought resistance and yield by enhancing photosynthesis and the antioxidant capacity of maize plants. *Frontiers in Plant Science* **9**:709 DOI 10.3389/fpls.2018.00709.

**Wang C, Lu G, Hao Y, Guo H, Guo Y, Zhao J, Cheng H. 2017.** *ABP9*, a maize bZIP transcription factor, enhances tolerance to salt and drought in transgenic cotton. *Planta* **246(3)**:453–469 DOI 10.1007/s00425-017-2704-x.

**Wang C-T, Ru J-N, Liu Y-W, Li M, Zhao D, Yang J-F, Fu J-D, Xu Z-S. 2018c.** Maize WRKY transcription factor *ZmWRKY106* confers drought and heat tolerance in transgenic plants. *International Journal of Molecular Sciences* **19(10)**:3046 DOI 10.3390/ijms19103046.

**Wang C-T, Ru J-N, Liu Y-W, Yang J-F, Li M, Xu Z-S, Fu J-D. 2018a.** The maize WRKY transcription factor *ZmWRKY40* confers drought resistance in transgenic *Arabidopsis*. *International Journal of Molecular Sciences* **19(9)**:2580 DOI 10.3390/ijms19092580.

**Wang H, Shao H, Tang X. 2016b.** Recent advances in utilizing transcription factors to improve plant abiotic stress tolerance by transgenic technology. *Frontiers in Plant Science* **7**:67 DOI 10.3389/fpls.2016.00067.

**Wang C, Shi X, Liu L, Li H, Ammiraju JSS, Kudrna DA, Xiong W, Wang H, Dai Z, Zheng Y, Lai J, Jin W, Messing J, Bennetzen JL, Wing RA, Luo M. 2013.** Genomic resources for gene discovery, functional genome annotation, and evolutionary studies of maize and its close relatives. *Genetics* **195(3)**:723–737 DOI 10.1534/genetics.113.157115.

**Wang Z, Su G, Li M, Ke Q, Kim SY, Li H, Huang J, Xu B, Deng X-P, Kwak S-S. 2016c.** Overexpressing *Arabidopsis* ABF3 increases tolerance to multiple abiotic stresses and reduces leaf size in alfalfa. *Plant Physiology and Biochemistry* **109**:199–208 DOI 10.1016/j.plaphy.2016.09.020.

**Wang W, Vinocur B, Shoseyov O, Altman A. 2004.** Role of plant heat-shock proteins and molecular chaperones in the abiotic stress response. *Trends in Plant Science* **9(5)**:244–252 DOI 10.1016/j.tplants.2004.03.006.

**Wang D, Yanchong Y, Liu Z, Shuo L, Wang Z, Xiang F. 2016a.** Membrane-bound NAC transcription factors in maize and their contribution to the oxidative stress response. *Plant Science* **250**:30–39 DOI 10.1016/j.plantsci.2016.05.019.

**Wang C-T, Yang QA, Yang Y-M. 2011.** Characterization of the *ZmDBP4* gene encoding a CRT/DRE-binding protein responsive to drought and cold stress in maize. *Acta Physiologiae Plantarum* **33(2)**:575–583 DOI 10.1007/s11738-010-0582-y.

**Wang B, Zheng J, Liu Y, Wang J, Wang G. 2012.** Cloning and characterization of the stress-induced bZIP gene *ZmbZIP60* from maize. *Molecular Biology Reports* **39(5)**:6319–6327 DOI 10.1007/s11033-012-1453-y.

**Waters AJ, Makarevitch I, Noshay J, Burghardt LT, Hirsch CN, Hirsch CD, Springer NM. 2017.** Natural variation for gene expression responses to abiotic stress in maize. *Plant Journal* **89(4)**:706–717 DOI 10.1111/tpj.13414.

**Wei K-F, Chen J, Chen Y-F, Wu L-J, Xie D-X. 2012b.** Molecular phylogenetic and expression analysis of the complete WRKY transcription factor family in maize. *DNA Research* **19(2)**:153–164 DOI 10.1093/dnares/dsr048.

**Wei K, Chen J, Wang Y, Chen Y, Chen S, Lin Y, Pan S, Zhong X. 2012a.** Genome-wide analysis of bZIP-encoding genes in maize. *DNA Research* **19(6)**:463–476 DOI 10.1093/dnares/dss026.

**Wei T, Deng K, Liu D, Gao Y, Liu Y, Yang M, Zhang L, Zheng X, Wang C, Song W, Chen C, Zhang Y. 2016.** Ectopic expression of DREB transcription factor, *AtDREB1A*, confers tolerance to drought in transgenic *Salvia miltiorrhiza*. *Plant and Cell Physiology* **57(8)**:1593–1609 DOI 10.1093/pcp/pcw084.

**Wei H, Zhao H, Su T, Bausewein A, Greiner S, Harms K, Rausch T. 2017.** Chicory R2R3-MYB transcription factors *CiMYB5* and *CiMYB3* regulate fructan 1-exohydrolase expression in response to abiotic stress and hormonal cues. *Journal of Experimental Botany* **68(15)**:4323–4338 DOI 10.1093/jxb/erx210.

**Wong CE, Li Y, Labbe A, Guevara D, Nuin P, Whitty B, Diaz C, Golding GB, Gray GR, Weretilnyk EA, Griffith M, Moffatt BA. 2006.** Transcriptional profiling implicates novel interactions between abiotic stress and hormonal responses in *Thellungiella*, a close relative of *Arabidopsis*. *Plant Physiology* **140(4)**:1437–1450 DOI 10.1104/pp.105.070508.

**Wu J, Zhou W, Gong X, Cheng B. 2016.** Expression of ZmHDZ4, a maize homeodomain-leucine zipper I gene, confers tolerance to drought stress in transgenic rice. *Plant Molecular Biology Reporter* **34(4)**:845–853 DOI 10.1007/s11105-015-0970-y.

**Wurzinger B, Mair A, Pfister B, Teige M. 2011.** Cross-talk of calcium-dependent protein kinase and MAP kinase signaling. *Plant Signal Behavior* **6(1)**:8–12 DOI 10.4161/psb.6.1.14012.

**Xiong L, Schumaker KS, Zhu JK. 2002.** Cell signaling during cold; drought; and salt stress. *Plant Cell* **14(suppl 1)**:S165–S183 DOI 10.1105/tpc.010278.

**Xu K, Xu X, Fukao T, Canlas P, Maghirang-Rodriguez R, Heuer S, Mackill DJ. 2006.** Sub1A is an ethylene-response-factor-like gene that confers submergence tolerance to rice. *Nature* **442(7103)**:705–708 DOI 10.1038/nature04920.

**Yamaguchi-Shinozaki K, Shinozaki K. 1994.** A novel cis-acting element in an *Arabidopsis* gene is involved in responsiveness to drought, low-temperature, or high-salt stress. *Plant Cell* **6(2)**:251–264 DOI 10.1105/tpc.6.2.251.

**Yamaguchi-Shinozaki K, Shinozaki K. 2006.** Transcriptional regulatory networks in cellular responses and tolerance to dehydration and cold stresses. *Annual Review of Plant Biology* **57(1)**:781–803 DOI 10.1146/annurev.arplant.57.032905.105444.

**Yamanouchi U, Yano M, Lin HX, Ashikari M, Yamada K. 2002.** Rice spotted leaf gene, *Spl7*, encodes a heat stress transcription factor protein. *Proceedings of the National Academy of Sciences of the United States of America* **99(11)**:7530–7535 DOI 10.1073/pnas.112209199.

**Yamasaki K, Kigawa T, Seki M, Shinozaki K, Yokoyama S. 2013.** DNA-binding domains of plant-specific transcription factors: structure, function, and evolution. *Trends in Plant Science* **18(5)**:267–276 DOI 10.1016/j.tplants.2012.09.001.

**Yan J, Warburton M, Crouch J. 2011.** Association mapping for enhancing maize (*Zea mays L.*) genetic improvement. *Crop Science* **51(2)**:433–449 DOI 10.2135/cropsci2010.04.0233.

Yanhui C, Xiaoyuan Y, Kun H, Meihua L, Jigang L, Zhaofeng G, Zhiqiang L, Yunfei Z, Xiaoxiao W, Xiaoming Q, Yunping S, Li Z, Xiaohui D, Jingchu L, Xing-Wang D, Zhangliang C, Hongya G, Li-Jia Q. 2006. The MYB transcription factor superfamily of *Arabidopsis*: expression analysis and phylogenetic comparison with the rice MYB family. *Plant Molecular Biology* **60(1)**:107–124 DOI 10.1007/s11103-005-2910-y.

Ying S, Zhang D-F, Fu J, Shi Y-S, Song Y-C, Wang T-Y, Li Y. 2012. Cloning and characterization of a maize bZIP transcription factor, *ZmbZIP72*, confers drought and salt tolerance in transgenic *Arabidopsis. Planta* **235(2)**:253–266 DOI 10.1007/s00425-011-1496-7.

Yu S, Liao F, Wang F, Wen W, Li J, Mei H, Luo L. 2012. Identification of rice transcription factors associated with drought tolerance using the Ecotilling method. *PLOS ONE* **7(2)**:e30765 DOI 10.1371/journal.pone.0030765.

Yu X, Liu Y, Wang S, Tao Y, Wang Z, Shu Y, Peng H, Mijiti A, Wang Z, Zhang H. 2016. *CarNAC4*, a NAC -type chickpea transcription factor conferring enhanced drought and salt stress tolerances in *Arabidopsis. Plant Cell Reports* **35(3)**:613–627 DOI 10.1007/s00299-015-1907-5.

Zhang Y, Ge F, Hou F, Sun W, Zheng Q, Zhang X, Ma L, Fu J, He X, Peng H, Pan G, Shen Y. 2017b. Transcription factors responding to Pb stress in maize. *Genes* **8(9)**:231 DOI 10.3390/genes8090231.

Zhang X, Liu X, Wu L, Yu G, Wang X, Ma H. 2015. The SsDREB transcription factor from the succulent halophyte *Suaeda salsa* enhances abiotic stress tolerance in transgenic tobacco. *International Journal of Genomics* **2015**:875497 DOI 10.1155/2015/875497.

Zhang X, Liu X, Zhang D, Tang H, Sun B, Li C, Hao L, Liu C, Li Y, Shi Y, Xie X, Song Y, Wang T, Li Y. 2017a. Genome-wide identification of gene expression in contrasting maize inbred lines under field drought conditions reveals the significance of transcription factors in drought tolerance. *PLOS ONE* **12(7)**:e0179477 DOI 10.1371/journal.pone.0179477.

Zhang X, Wang L, Meng H, Wen H, Fan Y, Zhao J. 2011. Maize *ABP9* enhances tolerance to multiple stresses in transgenic *Arabidopsis* by modulating ABA signaling and cellular levels of reactive oxygen species. *Plant Molecular Biology* **75(4–5)**:365–378 DOI 10.1007/s11103-011-9732-x.

Zhao Y, Ma Q, Jin X, Peng X, Liu J, Deng L, Sheng L. 2014. A novel maize homeodomain-leucine zipper (HD-Zip) I gene, *Zmhdz10*, positively regulates drought and salt tolerance in both rice and *Arabidopsis. Plant and Cell Physiology* **55(6)**:1142–1156 DOI 10.1093/pcp/pcu054.

Zhao X, Peng Y, Zhang J, Fang P, Wu B. 2018. Identification of QTLs and Meta-QTLs for seven agronomic traits in multiple maize populations under well-watered and water-stressed conditions. **58(2)**:507–520 DOI 10.2135/cropsci2016.12.0991.

Zhao Y, Zhou Y, Jiang H, Li X, Gan D, Peng X, Zhu S, Cheng B. 2011. Systematic analysis of sequences and expression patterns of drought-responsive members of the HD-Zip gene family in maize. *PLOS ONE* **6(12)**:e28488 DOI 10.1371/journal.pone.0028488.

Zhou W, Jia C, Wu X, Hu R, Yu G, Zhang X, Liu J, Pan H. 2016. *ZmDBF3*, a novel transcription factor from maize (*Zea mays L.*), is involved in multiple abiotic stress tolerance. *Plant Molecular Biology Reporter* **34(1)**:353–364 DOI 10.1007/s11105-015-0926-2.

Zhu JK. 2002. Salt and drought stress signal transduction in plants. *Annual Review of Plant Biology* **53**:247–273 DOI 10.1146/annurev.arplant.53.091401.143329.

Zhu Q, Zhang J, Gao X, Tong J, Xiao L, Li W, Zhang H. 2010. The *Arabidopsis* AP2/ERF transcription factor *RAP2.6* participates in ABA, salt and osmotic stress responses. *Gene* **457(1–2)**:1–12 DOI 10.1016/j.gene.2010.02.011.

Zhuang J, Deng D-X, Yao Q-H, Zhang J, Xiong F, Chen J-M, Xiong A-S. 2010. Discovery, phylogeny and expression patterns of AP2-like genes in maize. *Plant Growth Regulation* **62(1)**:51–58 DOI 10.1007/s10725-010-9484-7.