# Peer review of "Transcription factors involved in abiotic stress responses in Maize (Zea mays L.) and their roles in enhanced productivity in the post genomics era"

_PeerJ, doi:10.7717/peerj.7211_

## Round 0.1 · original submission · Minor Revisions

We received 3 reviews with complementary comments. Please consider the remarks, especially from the file attached by Reviewer 3. I believe the manuscript needs moderate revision and one more review round.

·

Basic reporting
* * *
Experimental design
* * *
Validity of the findings
* * *
Additional comments

Commend the authors for their extensive data set, compiled over of many researches. In addition the manuscript is clearly written in most parts. If there is a weakness, it is in the 1 and 2 which should be improved upon before Acceptance.
1-The English language should be improved to ensure that an International audience can clearly understand your text. Some examples where the language could be improved include lines 167, 454, 456– the current phrasing makes comprehension difficult.
In line 167(intricate) ~ it is possible instead of this word (complicated).
In line 454 subjugated ~ it is possible instead of this word (submit or subject).
In line 456 immense ~ it is possible instead of this word (enormous) and so on.
2-In Figure (1) Abbreviations meaning must be written below
For example:-
Abscisic acid (ABA).
Reactive oxygen species (ROS).
myeloblastosis oncogene (MYB).
myelocytomatosis oncogene (MYC)
ABA-responsive element binding protein (AREB). ABA-binding factor (ABF).
ABA- independent regulons include; the NAC (CUC, NAM and ATAF)
The zinc-finger homeodomain (ZF-HD) regulon
The dehydration responsive element binding proteins (DREBs)
The cis-acting element (DRE)
The cis-acting element (CRE)
C-repeat (CRT) and so on.

Thank you very much for this wonderful opportunity to participate in directing this work.

Dr. Salwa El-Sayed Morsy Mohamed.
Molecular Cell Biology Lecturer
Department of Molecular Biology
Genetic Engineering & Biotechnology Research Institute (GEBRI),
University of Sadat City-Sadat City, Egypt. P.O. Box 22857/79
Tel: +2 01095215769
Email: salwa3eg@yahoo.com
salwa.khedr@gebri.usc.edu.eg

·

Basic reporting

Langauge may be improved and typos need to be corrected.

Experimental design

Some of the paragraphs need to be restructured to have more clairity.

Validity of the findings

Authors need to add stress-specific role of TFs and the cross-talks of TFs during stress.

Additional comments

The review titled “Transcription Factors involved in Abiotic stress responses in Maize (Zea mays L.) and their roles in Enhanced Productivity in the Post Genomics Era” compiled various experiments and approaches. I appreciate the efforts put up by the authors.

Following are the major comments:
Introduction:
Line 85 to 105: I think you have to define the content of a paragraph. For example, this paragraph provides information about many subjects. Arrange something like “uses of maize in one paragraph (economical, genetical, functional, etc. uses), types of economically important abiotic stresses in maize in the next paragraph and types of different molecular mechanisms identified in maize in another paragraph”.
Line 106 to 112: Wondering why do you want focus only on ROS when several pathways are involved in stress tolerance? Better provide complete review of literature about different mechanisms.
Line 106 to 112: Please simplify the content.
Line 116 to 118: Please rewrite as there is no meaning in the sentence.

Transcription factors:
Please follow some pattern while compiling the experiments, like stress-wise or function-wise to have a better clarity on the role of TFs.
Line 157 to 237: The whole page is about TFs in different crops but not from maize except one or two references. Instead, you might want to provide a generic information on different types of TFs, number of TF identified in maize, structural characterisation etc.
From the TFs described in the review, we have come to know that they are involved in drought/salt/cold/etc tolerance. It would be nice to get to know the actual functional mechanisms of those TF in stress tolerance.
Line 743 to 747: “Li et al. (2018) analyzed the expression of ZmDREB4.1 and found that overexpression of this gene in transgenic tobacco led to repressed stem elongation and petiole, hypocotyl and leaf extension. In maize, Overexpression of this gene suppressed growth and regeneration of the calli. However, ZmDREB4.1 was not induced by any abiotic or biotic stress treatments.”

This para doesn’t add any value to the draft. I might suggest that avoid information that are not going to add any value to the draft.

Tables and figures/Additional information:
Additional tables explaining the functional and trait-specific role of TFs would provide useful information to the researchers. In the title, it is mentioned “productivity”. Are any TFs associated with agronomically important traits?
As you mentioned in the introduction, the role of TFs during stress is complex and the tolerance mechanisms involve cross-talks of many pathways. Adding figures to explain the cross-talk of TFs and explain their role in abiotic stress tolerance would be useful.
You also want split the Figure 1 for specific stresses rather providing generic and simplified overview of stress tolerant pathways.

Minor comments:
I could casually find a lot of typo with complex sentence formation. Please get a professional copy-editing support.
Line no 80: Change “yields” in to “yield”. Throughout the text use yield.
Line no 83: “at diverse times during the crop growth stages”. You mean to say “during the different crop growth stages”?
Line no 88: Change “This” in to “The”
Line no 89: Use current year statistics on yield.
Line 442: “Sevens” to “seven”

·

Basic reporting

1- The review is covered most published articles in the field of review and within the scope of the journal.

2- Based on my knowledge, there is no the same review in maize. I think this is a useful review.

3- In the introduction, the subject was clearly introduced.

Experimental design

1- The survey methodology section must be extended.

2- The sources are enough and quoted appreciate.

3- The review organized logically.

Validity of the findings

1- The introduction was prepared very well, and it appears that the authors used enough references and could make a logical relation between them and their conclusion is valid, but it is necessary the conclusion will be rewritten and cover their findings.

2- The review includes a useful section as “future outlook” that give a future direction.

Additional comments

Conclusion section must be rewritten and cover the findings.

---

## Round 0.2 · accepted · Accept

Thank you for the updates and manuscript resubmission. All the major concerns are solved now. We have only one review now, but other reviewers had no critical comments at first reviewing stage. I think the manuscript could be published in current form.

# ·

Basic reporting

Met all the criteria.

Experimental design

Met all the criteria.

Validity of the findings

Met all the criteria.

Additional comments

The authors addressed all the comments very well.